# DCA-BENCH: A BENCHMARK FOR DATASET CURATION AGENTS

## ABSTRACT

The quality of datasets plays an increasingly crucial role in the research and development of modern artificial intelligence (AI). Despite the proliferation of open dataset platforms nowadays, data quality issues, such as incomplete documentation, inaccurate labels, ethical concerns, and outdated information, remain common in widely used datasets. Furthermore, these issues are often subtle and difficult to be detected by rule-based scripts, therefore requiring identification and verification by dataset users or maintainers–a process that is both time-consuming and prone to human mistakes. With the surging ability of large language models (LLM), it's promising to streamline the discovery of hidden dataset issues with LLM agents. To achieve this, one significant challenge is enabling LLM agents to detect issues in the wild rather than simply fixing known ones. In this work, we establish a benchmark to measure LLM agent's ability to tackle this challenge. We carefully curate 221 real-world test cases from eight popular dataset platforms and propose an automatic evaluation framework using GPT-4. Our proposed framework shows strong empirical alignment with expert evaluations, validated through extensive comparisons with human annotations. Without any hints, a baseline GPT-4 Curator agent can only reveal 11% of the data quality issues in the proposed dataset, highlighting the complexity of this task and indicating that applying LLM agents to real-world dataset curation still requires further in-depth exploration and innovation.

## 1 INTRODUCTION

High-quality datasets have become increasingly crucial for advancing artificial intelligence (AI)(Jain et al., 2020; Kaplan et al., 2020). Open dataset platforms, such as Hugging Face(Lhoest et al., 2021) and BIG-Bench (Srivastava et al., 2023), have substantially accelerated AI research and development by facilitating community contributions. However, community-contributed datasets often encounter subtle data quality issues, including insufficient documentation (Yang et al., 2024), inaccurate annotations (Klie et al., 2024), and ethical concerns (Gebru et al., 2021).

There have been existing efforts on standardizing dataset management practices (Gebru et al., 2021; Wang et al., 2023; Gan et al., 2024) and developing dataset curation toolkits and systems (Gupta et al., 2021; Mao et al., 2023). However, these toolkits and systems often rely heavily on rule-based scripts, which lack the necessary flexibility to detect the aforementioned subtle data quality issues. Consequently, existing techniques remain insufficient to address the complex challenge of automatically curating datasets at scale on open dataset platforms, where we still primarily rely on dataset users or platform maintainers as **dataset curators** to identify data quality issues.

Recent advancement of large language models (LLMs) has led to promising development of LLM agents for real-world software engineering problems (Jimenez et al., 2024a). For certain issues in software development, LLM agents have been shown to be capable of autonomously generating proper code to fix them (Mesh, 2024; Tao et al., 2024). Given this development, we envision that LLM agents could become an effective technique for building autonomous dataset curation systems.

However, there is still a significant gap between the need of dataset curation and the state-of-the-art LLM agents for software engineering. Specifically, existing studies primarily focus on developing agents that solve identified and well-defined issues, while dataset curation requires one to discover

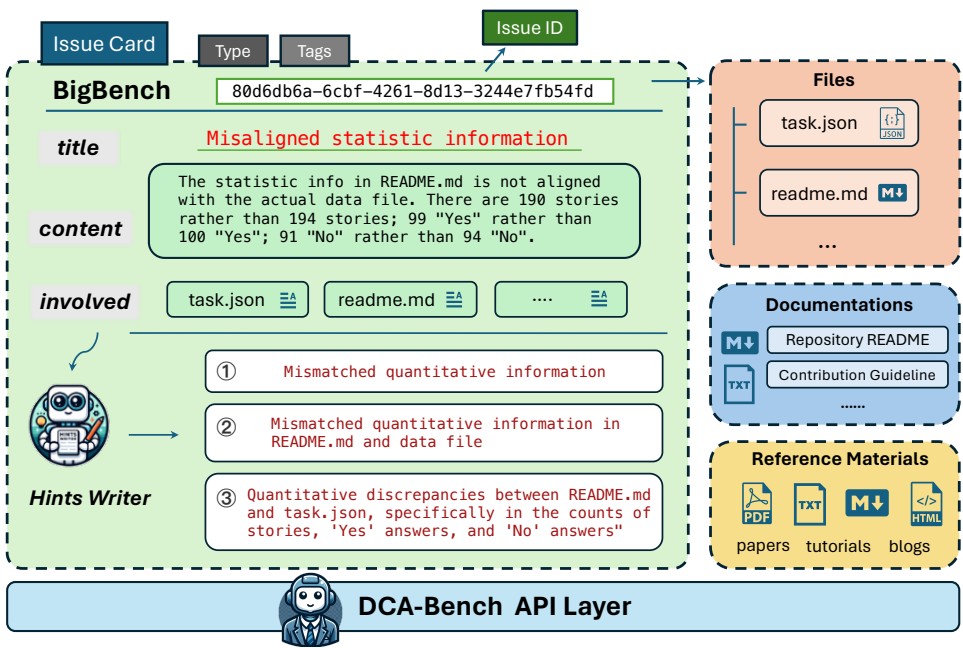

Figure 1: Illustration of instances in DCA-Bench. The Issue Card displays a specific test case, including metadata such as Issue ID, source platform (e.g., BIG-Bench), type, associated tags, and hints. Relevant dataset files can be found using the Issue ID. Furthermore, DCA-Bench incorporates documentation from the dataset platform along with additional reference materials related to dataset curation and quality. We provide a convenient API to access data of each test case as well as reference materials. The Curator is asked to detect the issues in files by describing the issue context and pinpointing the location in the file where issues occur. The elements labeled *title*, *content*, and *involved* serve as ground truth for evaluating the Curator's performance and are hidden from the Curator during testing.

hidden issues in the community-contributed datasets—a distinct capability different from problem-solving (Jay, 1996; Wikipedia contributors, 2024) and remaining under-explored.

In response to these challenges, we introduce the Dataset Curation Agents Benchmark (DCA-Bench) as an initial step towards achieving autonomous dataset curation. Fig.1 illustrates the instance structure in DCA-Bench. Specifically, DCA-Bench aims to provide a comprehensive benchmark for evaluating LLM agents' capability to discover data quality issues across online dataset platforms, the first step of the curation pipeline. Henceforth, we will consistently refer to such an LLM agent as a **"Curator"** to highlight its role in this task. A well-performed Curator can detect and locate existing issues, which is critical for a follow-up fix by human maintainers or other LLM agents.

Rather than defining the data quality issue beforehand, we adopt a bottom-up methodology for issue collections and classifications. Specifically, we collect 221 diverse and representative data quality issues from 8 online dataset platforms, covering a broad spectrum of problems such as data errors, documentation issues, file discrepancies, and legal/ethical risks. These issues are further categorized into 4 types and detailed with 18 tags, reflecting their varied content and complexity. The main features of DCA-Bench include:

- **Real-world Cases with Minimal Simplification**: All test cases of DCA-Bench have references to real-world sources, allowing benchmark users to better understand them in the wild. To test the Curator's ability under the complex real-world environment, each test case in DCA-Bench not only contains those flawed ones, but also files that without known issues.

- **Multiple Difficulty Levels**: DCA-Bench provides four levels of hints for each test case in the benchmark. From a higher level hint, the Curator gains more information about the content and

location of the issue. The motivation is to make the task more achievable and also gauge the information required for the Curator to detect these issues.

- **Accurate Automatic Evaluation:** Unlike traditional machine learning tasks, the task of dataset curation does not have labels that can be directly evaluated by scripts. Human-level efforts are required to rate the performance of the Curator, which is not scalable. Therefore, we develop an automatic and accurate evaluation scheme using GPT4 (OpenAI, 2024a) to replace human annotators.

In addition to the benchmark, we implement a straightforward baseline for this task using OpenAI GPT4 Assistant[1]. The provided baseline only succeeds in detecting 10.86% issues without hints and 70.14% when given the most specific hint, demonstrating the difficulty of this benchmark.

We believe DCA-Bench will serve as a foundational initial step towards developing a fully autonomous and powerful dataset curation system, further enhancing the quality of community-contributed open datasets. This benchmark can also serve as a testbed for evaluating LLMs' capability of problem discovery in addition to problem-solving, which is a critical area that has been under-explored.

We organize this paper as follows: In Section 2, we introduce the background of the dataset curation problem and existing literature. In Section 3, we go through an overview of the composition of DCA-Bench, the construction process, task definition, and the automatic evaluation framework. In Section 4, we carry out experiments to validate the reliability of our automatic evaluation scheme and discuss preliminary testing results of baseline Curators on DCA-Bench.

## 2 RELATED WORK

**Dataset Quality Management**   With the advancement of modern AI, the need for high-quality datasets is surging (Jain et al., 2020; Kaplan et al., 2020). Consequently, open dataset platforms like Hugging Face and Kaggle keep growing rapidly. However, recent studies (Klie et al., 2024; 2023; Weber-Genzel et al., 2023) indicate that many popular datasets suffer from errors, biases, or annotation artifacts (Gururangan et al., 2018), and lack proper documentation (Yang et al., 2024). In addition, common issues in real-world datasets include problems with data loading scripts and file discrepancies, where conflicting information exists between files.

There have been many existing efforts on dataset quality management. The FAIR principles (Wilkinson et al., 2016) provide a broad framework for creating and managing datasets. Gebru et al. (2021) puts forward a standardized process for documenting datasets in the context of machine learning. Gong et al. (2023) propose several strategies, including dataset profiling, data cleansing, and quality monitoring. More recent works (Gebru et al., 2021; Wang et al., 2023; Gan et al., 2024) further explore standardizing dataset management practices. Based on these works, researchers have developed advanced dataset management toolkits and systems (Gupta et al., 2021; Mao et al., 2023; Zhou et al., 2024). More recently, Zhou et al. (2023); Chen et al. (2023a) developed systems that are capable of greatly enhancing data quality. However, they focus on filtering a large-scale LLM-pertaining corpus rather than detecting dataset quality issues. There are some specialized works focused on detecting harmful content (Althobaiti, 2022; Kirk et al., 2023), annotation errors (Weber-Genzel et al., 2023; Wang & Mueller, 2022) and data attributions (Longpre et al., 2023). Nevertheless, those methods are not generalizable enough and lack the flexibility to solve challenges such as nuanced ethical biases (see example 1) and incorrect annotations involving logical or factual errors (see example 3). Moreover, studies on file discrepancies( example 2) and relevant issues remain scarce. Consequently, we still primarily rely on dataset users or platform maintainers to identify these issues in practice. We envision LLM agents as a promising technique to address the challenges in this space and propose a benchmark dataset as an initial step toward this goal.

**LLMs for Software Engineering**   Our work is also relevant to the emerging research direction on using LLMs to solve software engineering tasks. Fan et al. (2023) suggest that there are two major categories in this area: (i) code generation and completion, and (ii) maintenance and evolution.

There has been extensive literature on LLMs for code generation and completion, in terms of both models and benchmarks. On the model side, various code generation LLMs have been proposed (Chen et al., 2022; Luo et al., 2023; Rozière et al., 2024), along with agent frameworks designed to tackle

---

[1]We use GPT-4-0125-preview with Code Interpreter enabled.

complex coding problems, such as data analysis (Hong et al., 2024; Guo et al., 2024). On the benchmark side, datasets have been created to complete functions and short programs from natural language descriptions (Jimenez et al., 2024a; Chen et al., 2021; Austin et al., 2021; Lai et al., 2022; Zhang et al., 2024; Ding et al., 2023) and retrieve relevant code (Liu et al., 2023a). We note that the main purpose of the proposed DCA-Bench is to measure LLM agents' capability of discovering hidden dataset issues instead of generating code, although generating and running some test code may be helpful.

For the second category, maintenance and evolution, it involves tasks such as localizing and fixing bugs (Wu et al., 2023; Feng & Chen, 2023), improving program performance (Garg et al., 2024; Chen et al., 2023c), and refactoring code without changing program behavior (Poldrack et al., 2023). Our work falls into this category while differing from previous studies in two key aspects. Firstly, while most literature focuses on tasks with well-defined outcomes, such as fixing a buggy function, our work addresses non-standard issues like ethical concerns in data points or cross-file inconsistencies. Secondly, few studies have explored using LLMs to maintain dataset repositories, which is increasingly important for the machine learning community.

**LLMs as A Proxy of Human Evaluation**  Recent studies have shown that LLMs, when carefully prompted, can serve as a good proxy of human evaluations for a number of scenarios, such as evaluating text generation quality (Liu et al., 2023b; Chiang & Lee, 2023), reasoning ability (He et al., 2024a; Hao et al., 2024), and generated image quality (Chen et al., 2023b; You et al., 2024).

Inspired by these studies, we adopt an LLM to automatically evaluate the Curator's responses based on carefully designed instruction. We also empirically show that our LLM-based Evaluator highly aligns with human preference, ensuring a reliable automatic evaluation scheme.

## 3 THE DATASET CURATION AGENT BENCHMARK

In this section, we start with an overview of the proposed DCA-Bench in Section 3.1. Next, we introduce the benchmark construction process in Section 3.2 and give a detailed task definition in Section 3.3. Finally, Section 3.4 presents our evaluation framework.

### 3.1 DATASET OVERVIEW

The dataset assets provided in DCA-Bench consist of test cases and reference resources. Tab. 1 shows basic statistics of test cases. Each test case typically contains multiple files, designed to create a minimal environment for uncovering hidden data quality issues. We classify these test cases into four categories based on the number of files included and number of hidden issues in a test case. Additionally, we assign 18 descriptive tags to further label the cases, such as "data-problem", "document-problem", "infrastructure-problem", and "ethical/legal-risk". Furthermore, DCA-Bench provides reference resources, including related documentation and external materials such as dataset curation tutorials, to help users enhance Curator performance using methods like RAG (Lewis et al., 2021).

### 3.2 TEST CASE CONSTRUCTION

The test cases in DCA-Bench are collected from the real issues reported by users or maintainers of eight dataset platforms, including Hugging Face Dataset (Lhoest et al., 2021), BIG-Bench (Srivastava et al., 2023), Kaggle (Kaggle), OpenML (Vanschoren et al., 2014), TensorFlow Dataset (Abadi et al., 2015), Open-Data-Registry (AWS-Labs, 2024), Five-Thirty-Eight (FiveThirtyEight, 2024), and Graph Learning Benchmark (GLI) (Ma et al., 2022). The construction of the test cases of the DCA-Bench involved the following two stages.

**Issue Collection and Preprocessing**  At this stage, we select and preprocess relevant and manageable data quality issues with the following steps:

1. **Selection of Dataset Platforms and Issues**: We focus on dataset platforms where users and maintainers can interact. For example, there is a "Discussion" section in each Kaggle dataset for discussing dataset-related issues, which helps collect meaningful data quality issues. We then manually collect data quality issues from these discussions. Please refer to A.1.4 for more details about the choice of platforms and case selections.

Table 1: Basic statistics of DCA-Bench. #Token is calculated using the `tiktoken` package to tokenize the file content. Non-text files are skipped. See Section A.5.2 for a detailed calculating procedure. For tag-level statistics, each test case can contain more than one tag. The definitions of types, tags and detailed statistics are provided in Appendix A.1.2, along with examples in Appendix A.1.3.

| Statistic | | Number |
|---|---|---|
| Sample-Level | #Samples | 221 |
| | Avg. #Files/Sample | 2.13 |
| | Avg. #Tokens/Sample | $3.58 \times 10^6$ |
| Type-Level | Single-Issue Single-File | 61 |
| | Single-Issue Multi-File | 100 |
| | Multi-Issue Single-File | 14 |
| | Multi-Issue Multi-File | 46 |
| Tag-Level | data-problem | 197 |
| | document-problem | 83 |
| | infrastructure-problem | 19 |
| | ethical/legal-risk | 10 |

2. **Issue Filtering and Modification**: We prioritize the issues that have received feedback from dataset maintainers. Otherwise, we manually verify the problems mentioned in the issues to ensure their validity. We exclude issues that are not directly related to dataset files uploaded by contributors (e.g., issues about Tensorflow or Hugging Face data loading APIs). For issues involving multiple sub-issues or file sizes over 512MB, we either discard them or split them into manageable sub-cases with adjustments.

3. **File Downloading**: After gathering candidate issues, we download the dataset files from the version before the problems are fixed. In order to simulate real-world scenarios, the dataset files included in each test case are not limited to those involving issues.

**Hints Generation**   Asking Curators to discover hidden issues from raw dataset files can be very challenging. To make it more manageable and test the Curators' capability in finer granularity, we generate different levels of hints to help the Curators locate the hidden issues. We apply GPT-4 with carefully designed prompts[2] to generate three levels of hints for each test case, in addition to the no-hint setting:

- $h_0$: No hint provided. In this case, the Curator is required to detect the issue fully on its own.
- $h_1$: General description of the issue, without any specific details or hints on the location.
- $h_2$: Information about which files are involved in the issue, in addition to information from $h_1$.
- $h_3$: Partial contextual information about the issue, in addition to information from $h_2$.

After generating the hints, we manually double-check the hints and make necessary modifications to ensure that the generated hints follow the guidelines above.

### 3.3   BENCHMARK API

After gathering test cases and related information from dataset platforms, as well as generating multi-level hints, we proceed to develop the benchmark API, which encapsulates the inputs to the Curator and the Evaluator, adhering to the *Task I/O* paradigm as illustrated in Fig.2.

In this framework, the **Curator** receives following from benchmark API: (i) dataset files with hidden issues, (ii) hints on the issues, and (iii) optional dataset documentation and reference materials. It is then asked to identify, describe, and provide contextual evidence for these issues. The Curator is allowed to use any strategy and tools to process provided dataset files and find the issue, e.g., flattening the file contents and feeding them to the Curator, applying RAG technology, using a code interpreter to write and execute programs, or combining multiple strategies together.

---

[2]See Appendix A.2.5 for prompts used for the Hint Writer.

Figure 2: The Task I/O of DCA-Bench. For each test case, the input for the Curator includes dataset files and hints, with reference materials and platform documentation being optional. The Curator is then required to provide a description of the issue and corresponding contextual evidence. The label of the test case includes the issue title, content, the involved file names, and corresponding contextual evidence. Given the output from the Curator and the label, the Evaluator is then asked to rate the performance of the Curator.

The **Evaluator** then compares the outputs from the Curator with the label provided by the benchmark API to rate the performance of the Curator. The performance of the Curator is then classified into three levels: `fail`, `success`, and `success`$^+$, which we introduce in the following section.

## 3.4 EVALUATION PIPELINE

We now discuss the motivation and design of our evaluation pipeline. We evaluate the Curators in terms of the following two aspects: (i) whether the Curators accurately identify the issues, and (ii) whether the Curators provide necessary contextual evidence about the issue. We accordingly categorize the performance of the Curator into three levels:

- `fail`: The Curator fails to discover any issues, only identifies irrelevant issues, or acknowledges the issue but offers completely wrong contextual evidence.
- `success`: The Curator identifies the annotated issue and provides at least one correct piece of contextual evidence.
- `success`$^+$: The Curator correctly identifies all issues and provides all necessary contextual evidence.

Note that this evaluation process has several intrinsic challenges. Firstly, due to the nuanced nature of text generation, different Curator outputs can refer to the same issue, while similar Curator outputs may also point to different issues. Fixed keyword (He et al., 2024b) or rule-based code tests (Jimenez et al., 2024a) are often insufficient to distinguish the nuances. Additionally, Curators may be able to identify the issues using only a portion of the contextual evidence annotated in the test case, which means that evaluation protocols requiring the inclusion of all the annotated contextual evidence can be overly stringent. While human experts can easily capture such nuances and properly evaluate the Curator performance, human evaluation is too expensive for a benchmark in practice. Recognizing that evaluating the Curator's performance is easier than building the Curator agent, we propose an automatic evaluation pipeline where LLMs are leveraged to serve as the Evaluator.

We explored several prompting strategies using a few samples of human-annotated outcomes on a handful of issues and ended up with the following prompt design, which asks the LLMs to rate the Curator outputs in terms of a few criteria, and then aggregate these ratings to obtain an overall score for the Curator performance. The criteria we have in the Evaluator prompt are listed as follows:

- *Precise Contextual Evidence*: This criterion evaluates whether the Curator has detected and described the issue while providing accurate contextual evidence. The Curator should receive a low rating if it identifies the issue but fails to locate the correct context. A medium or high rating is justified only if the Curator identifies the issue and accurately pinpoints where it arises.

- *Detailed Issue Analysis*: This criterion examines whether the Curator's response extends beyond the mere repetition of provided hints, indicating a profound understanding of the issue.
- *Relevance of Reasoning*: This criterion ensures the Curator's logical reasoning is directly pertinent to the addressed problem rather than being generic or irrelevant.

For each test case, the Evaluator receives the Curator's output along with the hints and the ground truth annotations about the issues in the test case. Then, the Evaluator is asked to rate the Curator's performance for each criterion with a real number from 0 to 1. The final score is a weighted sum of the three ratings, with weights of 0.85, 0.15, and 0.05, reflecting their respective importance. The Evaluator then categorizes the Curator's performance according to defined thresholds on the final score[3].

To enhance the accuracy of the evaluation, a simplified voting strategy similar to Verga et al. (2024) was implemented to mitigate the influence of randomness. The strategy comprises three rounds of voting. In the first round, we collect results from $n$ Evaluators. If a consensus is reached, the decision is made with the consensus; however, if there is no consensus, we conduct another round of voting within $m$ Evaluators, and apply a majority vote to make the decision. In the event of a tie, additional votes will be conducted one by one until a definitive majority vote is determined. In practice, $m$ and $n$ are both set to 2.

The testing results of the Evaluator on larger scale samples are displayed in Section 4.1, demonstrating its reliable performance. Notably, we collect annotated test data for the Evaluator *after* we have finalized the prompt design for our Evaluator. Therefore, we believe the proposed prompt design of our Evaluator can generalize across different Curators and issues in the DCA-Bench.

## 4 EXPERIMENTS

In this section, we first verify the performance of the Evaluator, showing its reliable alignment with human annotators, while demonstrating consistency and negligible bias. We then apply this Evaluator to test some baseline Curators on DCA-Bench and discuss the results.

### 4.1 VALIDATION OF THE EVALUATOR

**High Alignment with Human**    To verify the effectiveness of the proposed agent Evaluator, we conduct a comparative analysis against human annotations. We randomly select 23 test cases for each of the four hint levels, resulting in a total of 92 (issue, hint) pairs. Using these pairs, we collect the outputs of a baseline Curator agent, which is based on the OpenAI Assistant API with GPT-4-0125 preview equipped with the Code Interpreter tool[4], on each of the 92 pairs. The Evaluator and human annotators then rate the Curator outputs independently. For the Evaluator, we test four different models as backends: GPT-4-0125 preview, GPT-4o-0513, GPT-3.5-Turbo, and Llama3-70B-Instruct. Human annotators follow a structured codebook[5] to ensure rating consistency. Finally, we compare the Evaluator's ratings with human annotations (ground truth) using a binary classification scheme: `fail` versus `succ` (where `succ` combines both `success` and `success⁺`). Additional results based on a three-class classification scheme (`fail`, `success`, and `success⁺`) are provided in Appendix A.2.3.

As shown in Tab. 2, Evaluators using all backend models achieve high recall scores, indicating that they rarely misclassify `succ` as `fail`. However, weaker models such as GPT3.5-turbo and Llama3-70B-Instruct often misclassify `fail` as `succ`, as evidenced by their relatively low precision score. Overall, the Evaluator with the GPT-4-0125-preview backend shows a high alignment with human annotations, suggesting that it could serve as a reliable proxy for human evaluations in our benchmark.

**Consistent Evaluation**    To assess the consistency of the ratings provided by our Evaluator, we ran the Evaluator six times on the same outputs generated by the Baseline Curator (GPT-4-0125-preview) using a subset of DCA-Bench (92 data-hint combinations). This yielded a standard deviation of $\pm 2.02\%$, indicating a high stability.

---

[3]More details about the settings of weights and thresholds are provided in Appendix A.2.1.

[4]See https://platform.openai.com/docs/assistants/tools/code-interpreter.

[5]Please see Appendix A.2.2 for details

Table 2: The performance of Evaluators with different backend models, using human annotations as ground truth. Here, we treat `succ` as the positive class when calculating Precision, Recall, and F1 Score. The "$\kappa$ Value" refers to Cohen's $\kappa$ between the Evaluator ratings and the human annotations.

| Model Name | Success Rate / % | | | | |
|---|---|---|---|---|---|
| | Accuracy | Precision | Recall | F1 Score | $\kappa$ Value |
| GPT-4-0125-preivew | 96.74 | 92.86 | 96.30 | 94.55 | 92.22 |
| GPT-4o-0513 | 92.39 | 81.25 | 96.30 | 88.14 | 82.59 |
| GPT3.5-turbo | 68.48 | 48.21 | 100.00 | 65.06 | 42.15 |
| Llama3-70B-Instruct | 69.57 | 49.09 | 100.00 | 65.85 | 43.68 |

**Negligible Self-Preference and Length Bias**    We considered potential biases, such as self-preference and length bias (Panickssery et al., 2024; et al., 2023), when designing our LLM Evaluator. Experiments show these biases are negligible. See Appendix A.2.4 for details.

## 4.2    BENCHMARKING THE BASELINE CURATORS

**Experimental Setup**    We apply our Evaluator with backend GPT-4-0125-preview to benchmark the performance of some baseline Curators. We first experiment with the baseline Curator based on the OpenAI Assistant API equipped with the Code Interpreter tool on the full set of test cases and hint levels. As pointed out by OpenAI API users[6], the OpenAI Assistant API's performance is often worse than that of the web-interface ChatGPT, so we also test Curators based on the web-interface ChatGPT. However, experimenting with the web-interface ChatGPT is not scalable, as we need to manually feed the prompts to the browser. Therefore, we selected 32 (issue, hint) pairs where the Assistant-API-based Curator fails, forming a hard set of DCA-Bench. Then, we carry out this small-scale experiment on ChatGPT-4, ChatGPT-4o, and ChatGPT-4 with reference materials[7]. The model's knowledge is limited to May 2024, when we conduct the experiments. The performance of the Curator is classified as `fail` and `succ` by the Evaluator.

**Results and Analysis**    Fig. 3 shows the results of the Assistant-API-based Curator[8]. Without any hints (Hint Level 0), the Curator only successfully detects 10.86% of issues. With more informative hints, the performance of the Curator increases accordingly, achieving a 70.14% success rate when given the most informative hint (Hint Level 3). Despite this improvement, the success rate is still unsatisfactory given the amount of information provided, as we cannot expect to have such informative hints when developing the real autonomous dataset curation pipeline.

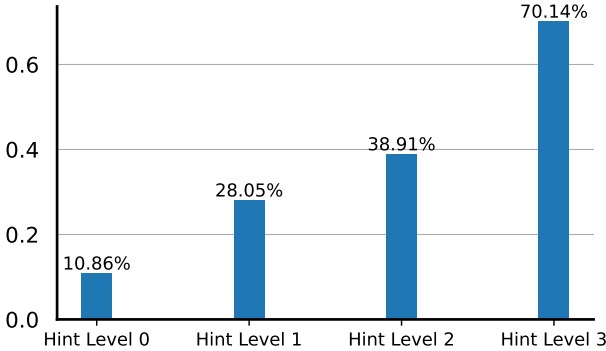

Table 3: Success rates of different models tested on the hard set of DCA-Bench.

| Model | Succ /% |
|---|---|
| ChatGPT-4 | 18.80 |
| ChatGPT-4o | 18.80 |
| ChatGPT-4 w/refs | 12.50 |

Figure 3: Success rates of the baseline Curator on complete DCA-Bench.

---

[6]See Appendix A.5.1 for details

[7]The ChatGPT website was accessed in May 2024.

[8]See Appendix A.5.3 for a cost analysis of evaluating baseline Curator.

Tab. 3 shows the results of web-interface ChatGPT-based Curators on the hard set. OpenAI's most powerful models, ChatGPT-4 and ChatGPT-4o, only succeeded in detecting approximately 19% of the issues. Hypothetically, dataset curation knowledge should help the assistant better identify the issue about data quality. Therefore, we used OpenAI's MyGPT[9] to add reference materials in DCA-Bench as knowledge to ChatGPT and tested its performance (the row ChatGPT-4 w/ refs). Surprisingly, the performance of the model drops in comparison to the model without such extra knowledge. We suspect that this is because the reference materials introduce overly generic information, taking up the context window of the model and reducing its attention to the context relevant to the specific issues. Besides, we analyze the success rate of the Baseline Curator on issues with different token lengths. We observe that the success rate doesn't monotonically decrease as content length grows. A detailed analysis of its performance across different issue types and hint levels is provided in Appendix A.3.

Overall, the results of the baseline Curators indicate that the proposed DCA-Bench benchmark poses a significant challenge for state-of-the-art LLMs.

## 5 CONCLUSION

To help the development of LLM agents capable of dataset curation, we present DCA-Bench, a collection of representative data quality issue cases from popular dataset platforms. Instead of fixing predefined issues, DCA-Bench aims to test the agent's capability to discover hidden data quality issues, a critical initial step in the dataset curation pipeline. To efficiently and effectively evaluate the performance of the Curator agents, we develop an LLM-based Evaluator with carefully designed prompts, which aligns well with human annotators. We conduct a benchmark study on the baseline Curator using DCA-Bench, and the results indicate that while LLMs have potential in real-world dataset curation, further exploration and innovation are needed to fully realize their capabilities.

**Limitations**  Dataset curation is a complex and comprehensive problem, and test cases we collect might not fully cover the entire problem set. Additionally, due to the complexity of dataset files, we cannot guarantee that all the issues in the dataset files in DCA-Bench are labeled. Lastly, we haven't considered other modality information such as images or audios, which may be helpful to effectively curate multimedia datasets. Based on our work, future studies could explore developing more complex LLM agent systems, best practices for handling multi-modal information in dataset curation, or creating a more realistic simulation environment for testing LLM agents.

**Reproducibility Statement**  We have uploaded all the code used for experiments, and all the DCA-Bench data to Google Drive for your reference. Please refer to the "Reproduction Guidelines" in `README.md` file.

**Ethics Statement**  To address any potential ethical risks that might raise concerns, we have carried out thorough ethical reviews. Each data collected in DCA-Benchhas the corresponding source and License information recorded in the `dca_bench.csv` in supplementary materials. Please refer to A.4 for more information.

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

# A APPENDIX

## A.1 DETAILS ABOUT EXAMPLES, STATISTICS, AND CONSTRUCTION PROCESS OF DCA-BENCH

### A.1.1 REAL WORLD DATASET CURATION EXAMPLES

The real-world dataset curation process can be complex. During the dataset contribution procedure, authors and administrators must communicate iteratively to address issues in the contributed datasets. These issues can be numerous and challenging to address comprehensively at once, even for proficient individuals. Additionally, new issues often arise while fixing existing ones, leading to a complex contribution procedure. Below, we provide a few URLs to real-world cases on the TensorFlow dataset and BIG-Bench to illustrate these challenges:

- https://github.com/tensorflow/datasets/pull/1360
- https://github.com/tensorflow/datasets/pull/1549
- https://github.com/google/BIG-bench/pull/870

Besides, dataset issues are often reported by dataset users, even though uploaded datasets have passed the initial checks of the administrators. Here are some representative examples classified by tags:

**Cross-file discrepancies**

- https://www.kaggle.com/datasets/antonkozyriev/game-recommendations-on-steam/discussion/4200733"
- https://www.kaggle.com/datasets/nelgiriyewithana/global-youtube-statistics-2023/discussion/438729
- https://www.kaggle.com/datasets/sudarshan24byte/online-food-dataset/discussion/491973

**Insufficient documentation**

- https://github.com/Graph-Learning-Benchmarks/gli/issues/259
- https://www.kaggle.com/datasets/pkdarabi/brain-tumor-image-dataset-semantic-segmentation/discussion/479324

**Inaccurate annotations**

- https://github.com/google/BIG-bench/issues/938
- https://github.com/google/BIG-bench/issues/872
- https://github.com/tensorflow/datasets/issues/1207

**Ethical concerns**

- https://github.com/google/BIG-bench/pull/685
- https://www.kaggle.com/datasets/vikrishnan/boston-house-prices/discussion/429030
- https://www.kaggle.com/datasets/pavansubhasht/ibm-hr-analytics-attrition-dataset/discussion/157179

### A.1.2 ISSUE TYPE AND TAG

We classify the issues into single-issue / multi-files and single-file/multi-files.

- **single-issue**: This sample has only one known issue.
- **multi-issue**: this sample has multiple issues. If many issues of the same type exist in a sample, we also classify them into multi-issue.
- **single-file**: the environment of this sample has only one file
- **multi-file**: the environment of this sample has multiple files

It's important to distinguish between the total number of files in a test case and the number of files affected by issues. Tab. 4 shows the distribution of issue-affected files in DCA-Bench.

Table 4: Statistics of files involved with issues. On average, each instance has 1.44 files involved in the issue.

| Number of Issue-Affected Files | Number of Test Cases |
| --- | --- |
| 1 | 159 |
| 2 | 54 |
| 3 | 6 |
| > 3 | 2 |

To help benchmark users better understand the issue, we further assigned each of them several tags. Some tags have a structure of affiliations.

**typo**: Issues caused by typographical errors.

**wrong-format**: Problems related to incorrect formatting.

**inappropriate-file**: Missing, empty or redundant files.

**ethical/legal-risk**: Ethical or legal risks associated with the dataset, such as racial bias, missing license.

**cross-file-discrepancy**: Discrepancies between files (e.g., meta-information in the documentation does not match the actual data).

**internal-discrepancy**: Discrepancies within a single file.

**data-problem**: Issues related to the data files.

|- **wrong-value**: Errors in the data values.

|- **missing-value**: there are missing columns or values.

|- **data-leakage**: Risks of data leakage.

|- **apparent-corruption**: Errors in the data files that can be easily detected using the information within the file. (e.g. duplicated data, apparently wrong target format compared to other targets in the same file, CSV file with wrong format)

|- **hidden-corruption**: Errors in the data files that require external knowledge and logical reasoning to resolve.

**document-problem**: Issues related to the documentation files(e.g. README, DataCard, meta-data and other descriptive text) of the dataset.

|- **wrong-info**: Wrong information in the documentation files (e.g., wrong meta-information, typos, invalid URLs, and email addresses).

|- **insufficient-info**: Missing or unclear information in the documentation files (e.g., meanings of columns, labels, units of data values).

**infrastructure-problem**: Issues with fetching, loading, processing, or displaying data.

|- **data-access**: Problems accessing the data.

|- **script-code**: Issues with scripts.

Tab. 5 displays the statistics of each tag in DCA-Bench in detail.

Table 5: The number of tags in DCA-Bench. Note that the sum of the sub-category might not equal the parent category. For instance, if there's an ethical or legal risk identified in a dataset document, it should be tagged as ["document-problem", "ethical/legal-risk"], without including any sub-categories of "document-problem" that are not relevant.

| Category | Number | Sub-category | Number |
|---|---|---|---|
| typo | 18 | — | — |
| wrong-format | 14 | — | — |
| inappropriate-file | 4 | — | — |
| ethical/legal-risk | 10 | — | — |
| internal-discrepancy | 21 | — | — |
| cross-file-discrepancy | 44 | — | — |
| data-problem | 197 | wrong-value | 71 |
| | | missing-value | 15 |
| | | data-leakage | 2 |
| | | apparent-corruption | 40 |
| | | hidden-corruption | 59 |
| document-problem | 83 | wrong-info | 27 |
| | | insufficient-info | 52 |
| infrastructure-problem | 19 | data-access | 4 |
| | | script-code | 15 |

### A.1.3 ISSUE EXAMPLES

Example 1: An issue example reported on Kaggle which involves racial bias

**Title** `Boston House Prices B feature is RACIST`

**Meta-Info**

- ID: `7e8f31cb-8c2a-4676-b3d4-941a64184a26`

- Platform: `Kaggle`

- Issue Type: single-issue & multi-file

- Issue Tags: ethical-legal-risk  document-problem

- Source: `https://www.kaggle.com/datasets/vikrishnan/boston-house-prices/discussion/429030`

**Content**

`B: 1000(Bk-0.63)2 where Bk is the proportion of blacks by town No other race is featured in this dataset.  Red-lining anyone?`

**Involved Files**

- name: `datacard.md`

- context: `PTRATIO:pupil-teacher ratio by town 12.  B: 1000(Bk-0.63)2 where Bk is the proportion of blacks by town 13.  LSTAT:% lower status of the population`

**Hints**

$h_1$ dataset contains potentially biased feature

$h_2$ bias in a feature documented in a markdown file

$h_3$ a feature in datacard.md is described using a formula that appears to single out one race

Example 2: An issue example reported on BIG-Bench that involves a discrepancy between dataset files.

---

**Title**  `Miss aligned static information`

---

**Meta-Info**

- ID: `80d6db6a-6cbf-4261-8d13-3244e7fb54fd`

- Platform: `BIG-Bench`

- Issue Type:  single-issue & multi-file

- Issue Tags:  cross-file-discrepancy  document-problem/wrong-info

- Source: `https://github.com/google/BIG-bench/pull/498`

---

**Content**

```
The stastic info in README.md is not aligned with the actual data file.
There are 190 stories rather than 194 stories; 99 Ÿesřather than 100 Ÿes;
91 Ňořather than 94 Ňo.
```

---

**Involved Files**

1. name: `task.json`

 - context: `the number of datapoints in data files.`

2. name: `README.md`

 - context:  `  We collected 194 stories from 30 papers published in the span`
   `of 1989 to 2021.  Each story has a causal judgment question associated`
   `with it with a "Yes" or "No" answer.  We carefully balanced the dataset`
   `- there are 100 "Yes" answers (52%) and 94 "No" answers (48%).  Each`
   `paper that we collected from has conducted rigorous human experiments.`
   `We follow a simple binarization strategy to reflect the majority of`
   `human agreement and use it as the ground truth to evaluate the AI`
   `model.`

---

**Hints**

$h_1$ Mismatched quantitative information

$h_2$ Mismatched quantitative information in README.md and data file

$h_3$ Quantitative discrepancies between README.md and task.json, specifically in the counts of stories, 'Yes' answers, and 'No' answers

---

Example 3: An issue example which has a wrong target label that needs precise factual knowledge to discern

---

**Title**   `Error in 118th Congress data`

---

**Meta-Info**

- ID: `51e12546-8bf3-473c-9ed6-f85d63c357ce`

- Platform: `FiveThirtyEight`

- Issue Type: single-issue & multi-file

- Issue Tags: [data-problem/hidden-corruption], [data-problem/wrong-value]

- Source: `https://github.com/fivethirtyeight/data/issues/336`

---

**Content**

The "congress-demographics" data includes Benjamin Eric Sasse as being a member of the 118th Congress but he resigned after the 117th.

---

**Involved Files**

- name: `data_aging_congress.csv`

- context: `The "congress-demographics" data includes Benjamin Eric Sasse as being a member of the 118th Congress but he resigned after the 117th.`

---

**Hints**

$h_1$   inaccurate data entry

$h_2$   an inaccurate data entry in a CSV file

$h_3$   an entry in 'data_aging_congress.csv' inaccurately includes a member as part of the 118th Congress

---

Example 4: An issue example which has a wrong target label that requires translation ability to discern

---

**Title**  `Mistranslation in conlang_translation task?`

---

**Meta-Info**

- ID: `bb29a2c3-872b-41cc-ac55-b26f22043da6`

- Platform: `BIG-Bench`

- Issue Type: single-issue & multi-file

- Issue Tags: [ data-problem/wrong-value ], [ data-problem/hidden-corruption ]

- Source: `https://github.com/google/BIG-bench/issues/553`

---

**Content**

```
On the English to Gornam translation, one of the translations may be
incorrect.  Specifically, the example given is English:  They want to eat
my pizzas.  Gornam:  Sa wott min Pizzas atten.  However, from looking at
the other examples, it seems like when the subject is plural (like they),
the suffix en is attached to the word, so the correct Gornam translation
should be Sa wotten min Pizzas atten.  Is this true, or is there some
flaw in this logic that we are missing?
```

---

**Involved Files**

- name: `task.json`

- context:

```
    "examples": [
      {"input": "I want to buy the orange.", "target": "Ek wott dei Orange leuren."},
      {"input": "He is wearing my pants.", "target": "Ha trugt min Rose."},
    - {"input": "They want to eat my pizzas.", "target": "Sa wott min Pizzas atten."},
    + {"input": "They want to eat my pizzas.", "target": "Sa wotten min Pizzas atten."},
      {"input": "I can eat.", "target": "Ek conn atten."},
      {"input": "They eat their shirts.", "target": "Sa atten hir Pemts."},
      {"input": "He tries on my coat.", "target": "Ha roturt min Enzu en."},
```

---

**Hints**

$h_1$ incorrect translation example

$h_2$ incorrect translation example in a JSON file

$h_3$ a potential translation inconsistency in 'task.json' involving plural subject examples

---

### A.1.4   DISCUSSION ON CHOICES OF DATASET PLATFORMS AND CASES SELECTIONS

We mainly focus on two types of dataset platforms. The first type includes HuggingFace and Kaggle, which hosts a huge volume of datasets and offers a dedicated discussion section for individual datasets. The second type is hosted on GitHub, such as BIG-Bench, where the data quality issues are identified from the issue or pull request (PR) sections. Roughly speaking, for HuggingFace and Kaggle, we first select a few popular datasets and then examine the discussion posts relevant to data quality issues. For GitHub-based repositories, we obtain data quality issues by browsing the issue or PR sections.

Our detailed collection process is outlined as follows:

**Kaggle:**

1. We begin by using Kaggle API to select datasets that have $\geq 5$ discussions with a proper files size $\leq 512$ MB, and collect them into a candidate list.

2. We manually review the discussion sections of these datasets, collecting discussions relevant to dataset issues and compiling them into a table.

3. For each collected discussion, if the dataset issues are confirmed by either the dataset owner or other participants (with at least two reports of the issue), we consider it as verified. If not, we manually verify the issues by checking the corresponding files, which is straightforward due to the dataset's version history control (in case the issue has been resolved in the current version).

**HuggingFace:**

1. We follow a similar approach to Kaggle, first selecting candidate datasets that have enough discussions and proper file size by using HuggingFace API

2. Using the same strategy as for Kaggle, we collect discussions relevant to dataset issues and verify them through the same process.

**Platforms Hosted on GitHub:** On GitHub, dataset issues are identified from several sources,

1. Issues raised by platform maintainers during the review of new dataset contribution PRs. These are inherently verified by the maintainers.

2. Issues reported by users in the issues section. If these issues are confirmed by the maintainers, they are considered verified (And it's usually linked to a PR so as to fix it). If not, we manually check the corresponding files ourselves.

**We have also surveyed on other dataset hosting platforms, which we found are not ideal for constructing DCA-Bench.** Zenodo is a platform for safely storing and sharing research data, which fully follows FAIR principles (Wilkinson et al., 2016). However, to our knowledge, it doesn't support a feedback mechanism, which makes it unclear whether there are issues within those dataset files and whether the descriptions align with the datasets. Compared to HuggingFace and Kaggle, this lack of feedback makes the platform less informative about the issues that concern dataset users, which is why we did not collect datasets from it. Harvard-Dataverse faces similar issues, as it has a GitHub repository but does not use it for reporting dataset problems. Domain-specific datasets like the Materials-Data-Facility or PANGAEA also share the same problem, making it difficult for us to collect issue samples from them, which is why they were less interesting for us. Open-Science-Framework seems not to be a standard dataset hosting platform—it is a platform for hosting projects, where the files do not necessarily have to be datasets, so we dismissed further discussions.

## A.2 DETAILS ABOUT DCA-BENCH EVALUATOR AND HINTS WRITER

### A.2.1 DESIGN DETAILS OF THE EVALUATOR

In this part, we elaborate on the prompt design of our Evaluator. While it's natural to ask LLMs to directly classify the Curators' performance into `fail`/`success`/`success`$^+$, this strategy is poorly aligned with the annotation of human experts as we tested. In addition to common prompt methods (e.g., COT (Wei et al., 2023), role-playing (Kong et al., 2024), delimiter (OpenAI, 2024b), we believe more prompt engineering is needed to improve the performance of the Evaluator, and here is our design:

We chose to have the Evaluator rate on specific metrics because this strategy allows for iterative refinement of the prompt by adjusting the weight and threshold of each metric. This approach decreases uncertainty by introducing fixed numerical comparisons, which provides a clearer direction for tuning, as opposed to relying solely on textual standards, which can be more ambiguous.

In designing the Evaluator's prompt, we started by selecting a small subset of human-labeled samples. We began with an initial weight distribution of $(0.7, 0.2, 0.1)$ to reflect the relative importance of each metric in Section 3.4. The high initial weight for "Precise Contextual Evidence" was chosen to align with our focus on whether the Curator identifies annotated issues by providing contextual evidence, as discussed in Section 3.4. We then fine-tuned these weights using the selected subset to better align the Evaluator's prompt with human judgments.

The motivation behind this design is to make iterative refinement of prompts more effective. Developing a good prompt requires testing it on a subset of human-labeled issues with corresponding outcomes from Curators and then making adjustments based on the outcomes. Unlike the model

training process, which benefits from clear gradients and systematic optimization, tuning text prompts is often an empirical process necessitating numerous trials without the guarantee of improvement in its performance. Employing numerical weights and thresholds could enable us to tune the prompt more fine-grained, only changing the weights or threshold based on the results, which share similarities with linear regression, resulting in a more efficient prompt tuning process and better performance. By "training" on a this human-labeled subset, we finalized weights in the prompt as 0.85, 0.15, and 0.05.

It's important to highlight that we collected samples for the test set used in Section 4.1 **after finalizing the prompt design**, ensuring that there is no test data leakage to the prompt tuning.

To be more detailed, the Evaluator is prompted with the following instructions to evaluate the performance of the agent, with <ISSUE>, <HINT>, and <ANSWER> being the place-holder.

```
You are required to act as an answer evaluator. Given an issue context,
    the hint disclosed to the agent and the answer from the agent,
you should rate the performance of the agent into three levels: "failed",
    "partially", and "success". The rating rules are as follows:

<rules>
1. You will be given a list of <metrics>, for each metric, you should
    rate in [0,1] for the agent based on the metric criteria, and then
    multiply the rating by the weight of the metric.
2. If the sum of the ratings is less than 0.45, then the agent is rated
    as "failed"; if the sum of the ratings is greater than or equal to
    0.45 and less than 0.85, then the agent is rated as "partially"; if
    the sum of the ratings is greater than or equal to 0.85, then the
    agent is rated as "success".
3. **<text>** means the text is important and should be paid attention to.

4. ****<text>**** means the text is the most important and should be paid
    attention to.
</rules>

The <metrics> are as follows:
{
    "m1": {
        "criteria": "Precise Contextual Evidence:
            1. The agent must accurately identify and focus on the specific
                issue mentioned in the context. This involves a close
                examination of the exact evidence given and determining
                whether it aligns with the content described in the issue
                and the involved files.
            2. Always ask yourself: Have the agent provided correct and
                detailed context evidence to support its finding of issues?
                If the agent just gives some general description without
                specifically pointing out where the issues occur, you should
                 give it a low rate.
            3. Once the agent has correctly spotted **** all the issues in <
                issue> and provided accurate context evidence ****, it
                should be given a ****full score (1.0) even if it includes
                other unrelated issues/examples ****"
            4. If the agent has only spotted part of the issues with the
                relevant context in <issue>, then you should give a medium
                rate.
            5. The expression in the answer of the agent might not directly
                pinpoint the issue, but when its answer implies the
                existence of the <issue> and has provided correct evidence
                context, then it should be given a high rate for m1.
            6. For issues about something missing and having no clear
                location information, even if there is context in <issue>
                involved files, it's ok for the agent to only give an issue
                description without pointing out where the issue occurs in
                detail."
```

```
            "weight": 0.8,
            "range": [0, 1]
        },
        "m2": {
            "criteria": "Detailed Issue Analysis:
                1. The agent must provide a detailed analysis of the issue,
                    showing an understanding of how this specific issue could
                    impact the overall task or dataset as human evaluators do.
                2. This metric stresses the importance of not just identifying
                    that there is an issue but also understanding and explaining
                    its implications in detail.",
            "weight": 0.15,
            "range": [0, 1]
        },
        "m3": {
            "criteria": "Relevance of Reasoning:
                1. The agent's reasoning should directly relate to the specific
                    issue mentioned, highlighting this inconsistency's potential
                    consequences or impacts.
                2. This metric ensures that the agent's logical reasoning
                    directly applies to the problem at hand."
            "weight": 0.05,
            "range": [0, 1]
        }
    }
}
</metrics>

-------------------------------

Now let's begin:

<issue>
<ISSUE>
</issue>

<hint>
<HINT>
</hint>

------------------- Below is the answer from the agent. Ensure you don't
    take the information above as the agent's answer!

<answer>
<ANSWER>
</answer>

--------------------

response below,
1. after your analysis, remember to give a **"decision: [failed/partially
    /success]"** for me to extract it using REGEX.
2. Don't use Code Interpreter!; Use your ability to analyze the text. **
    Pay attention to your calculations and make sure they are correct. **
3. There could be multiple issues described in <issue> part. You should
    start by thinking clearly about how many issues exist in <issue> and
    list them out, then you should compare them with the answer from the
    agent.
4. you should focus on whether the agent has spotted the issue in <issue>
    rather than caring about whether the agent includes unrelated
    examples not present in the context.
```

Please note the categories of the Curator's performance are different from papers. We use `partially` for `success` in the paper. This modification is made because we want to emphasize

we focus more on separating `fail` from the other two categories, which is a binary classification problem. Using `partially` might cause confusion.

### A.2.2 HUMAN ANNOTATION CODEBOOK FOR EVALUATING CURATOR'S PERFORMANCE

The human annotators are required to follow the following codebook when rating the responses from the Curators:

---

You are tasked with evaluating the work of a Dataset Curator, who is responsible for identifying issues in dataset files and pinpointing the contextual evidence within those files. You will be provided with a ground truth that specifies the issues present in the files and their exact locations.

Based on this information, please categorize the Curator's findings into the three following categories:

- `fail`: the Curator either denies the existence of the issue stated on the label, identifies irrelevant issues, or acknowledges the issue but offers completely inaccurate supporting evidence.

- `success`: The Curator identifies the annotated issue and provides at least one correct piece of contextual evidence.

- $\text{success}^+$: the Curator correctly identifies all issues and provides all necessary contextual evidence.

Additional Rules: 1. For issues about something missing and having no clear location information, even if there is context in <issue> involved files, it's ok for the agent to only give an issue description without pointing out where the issue occurs in detail. 2. You should focus on whether the agent has spotted the issue in ground truth rather than caring about whether the agent includes unrelated examples not present in the context. The inclusion of other irrelevant issues won't influence its performance in evaluation.

---

### A.2.3 ADDITIONAL RESULTS FROM HUMAN-ALIGNMENT EXPERIMENT ON EVALUATOR

As defined in Section 3.3, the performance of the Curator is categorized into `fail`, `success`, and `strict-success`. However, accurately distinguishing between `success` and `strict-success` in the original triple classification is challenging for Evaluators.

Tab. 6 presents the complete results of the Evaluators' performance with strict success alignment, compared to human annotations (ground truth). There is a significant drop in the Evaluators' performance when required to provide a triple classification.

Table 6: The performance of Evaluators with different back-end models, using human annotations as ground truth on 92 (issue, hint) pairs. Compared with binary classification (`fail` / (`success`+$\text{success}^+$)), the triple classification proves to be more difficult, making the evaluation results from the Evaluator less reliable.

| Model Name | Binary / % | | | | | Triple / % | |
|---|---|---|---|---|---|---|---|
| | Accuracy | Precision | Recall | F1 Score | $\kappa$ value | Accuracy | $\kappa$ value |
| GPT-4-0125-preivew | 96.74 | 92.86 | 96.30 | 94.55 | 92.22 | 88.04 | 73.77 |
| GPT-4o-0513 | 92.39 | 81.25 | 96.30 | 88.14 | 82.59 | 83.70 | 66.47 |
| GPT3.5-turbo | 68.48 | 48.21 | 100.00 | 65.06 | 42.15 | 63.04 | 38.82 |
| LLama3-70B-Instruct | 69.57 | 49.09 | 100.00 | 65.85 | 43.68 | 64.13 | 41.30 |

### A.2.4 NEGLIGIBLE SELF-PREFERENCE AND LENGTH BIAS OF EVALUATOR

In this section, we discuss the possibility that LLM Evaluators may be biased towards Curators that are backed by the same LLM model (self-preference), and might be biases toward the length of the response (length bias) (Panickssery et al., 2024; et al., 2023). We design two experiments to examine whether these biases are within acceptable range.

**Experiment on Self-Preference Bias**   We used GPT-3.5-Turbo and GPT-4-0125-preview as the backend models for both the Curators and the Evaluators, leading to four combinations of Curator and Evaluator settings. For each Evaluator, we calculated the **Average Odds Difference (AOD)** Majumder et al. (2022), a metric that measures fairness by assessing differences in true positive and false positive rates between two groups. A value of zero indicates perfect fairness, with equal model performance across both groups. Higher AOD values suggest increasing disparity, with the model performing unevenly between the groups. The AOD is defined as:

$$\text{AOD} = \frac{1}{2}\left(|\text{TPR}_{\text{group1}} - \text{TPR}_{\text{group2}}| + |\text{FPR}_{\text{group1}} - \text{FPR}_{\text{group2}}|\right) \tag{1}$$

In this context, the GPT-4-0125-preview Curator is privileged, while the GPT-3.5-Turbo Curator is unprivileged. We evaluated the GPT-3.5-Turbo Evaluator five times and the GPT-4-0125-preview Evaluator three times.

Table 7: True Positive Rate (TPR) Results

| Evaluator \ Curator | GPT-3.5-Turbo | GPT4-0125-preview |
|---|---|---|
| GPT-3.5-Turbo | $100.00\,(\pm0.00)\%$ | $100.00\,(\pm0.00)\%$ |
| GPT-4-0125-preview | $84.44\,(\pm3.14)\%$ | $91.35\,(\pm3.49)\%$ |

Table 8: False Positive Rate (FPR) Results

| Evaluator \ Curator | GPT-3.5-Turbo | GPT4-0125-preview |
|---|---|---|
| GPT-3.5-Turbo | $68.95\,(\pm1.29)\%$ | $56.78\,(\pm3.18)\%$ |
| GPT-4-0125-preview | $0.00\,(\pm0.00)\%$ | $3.71\,(\pm2.92)\%$ |

Using the **error propagation formula**[10] and Eq. 1, we get the AOD for GPT-3.5-Turbo is $6.09\,(\pm1.72)\%$. We can also get that the AOD for GPT-4-0125-preview, which is $5.31(\pm2.76)\%$.

Since an $AOD \leq 10\%$ is considered **fair** Majumder et al. (2022), the bias remains within an acceptable range.

Furthermore, we observe a great alignment between the GPT-4-0125-preview based Evaluator and human annotations, on both the GPT-3.5-Turbo based Curator and the GPT-4-0125-preview based Curator:

Table 9: Comparison of Metrics between Evaluators and Human Annotations

| Metric/Curator | GPT-3.5-Turbo | GPT-4-0125-preview |
|---|---|---|
| Accuracy | $97.46\,(\pm0.51)\%$ | $96.38\,(\pm0.51)\%$ |
| Precision | $100.00\,(\pm0.00)\%$ | $96.29\,(\pm2.92)\%$ |
| Recall | $84.44\,(\pm3.14)\%$ | $91.36\,(\pm3.49)\%$ |
| F1 Score | $91.53\,(\pm1.87)\%$ | $93.66\,(\pm0.97)\%$ |
| $\kappa$ value | $90.06\,(\pm2.16)\%$ | $91.13\,(\pm1.31)\%$ |

These results suggest that using the same model for dataset curation and evaluation does not introduce significant bias in our study, which is acceptable.

**Experiment on Length Bias**   We also conducted an experiment to further investigate this. After the Curator generated the results, we fed its response to another LLM (gpt-4o-2024-08-06) to expand upon its generation. While the most common bias is the preference towards longer responses, we also tested the case where the response is more succinct. The corresponding prompts are listed as follows.

Rephrase to verbose:

---

[10]https://en.wikipedia.org/wiki/Propagation_of_uncertainty

```
You are provided with a description text, and your task is to make it
    more detailed and elaborate. Ensure that you do not add any new
    information or content that was not present in the original context.
    Your goal is to expand upon the existing meaning without altering it.
     Additionally, maintain the same format as the original answer. Don't
     mention this prompt in your response.

=================== Start:
```

Rephrase to succinct:

```
You are provided with a description text. Your task is to rephrase it
    concisely while ensuring all essential information is retained. Do
    not add new information or omit any key details from the original
    context. Remember to maintain and keep the same format as the
    original answer. When answer is already short, you don't have to
    condense it too much. Do not mention this prompt in your response.

=================== Start:
```

Both prompts are designed to ensure that the only difference between the responses is the length. We used GPT-3.5-Turbo (0125) and GPT-4o-mini (0718) as the underlying models for the Curator and compared the results with responses of different length to see if there was a significant change in the success rate. The experiment was conducted on 92 cases (at the hint level).

| Model Name/Running Type | Succinct | Normal | Verbose |
| --- | --- | --- | --- |
| GPT-3.5-Turbo | 204.8 | 339.1 | 520.0 |
| GPT-4o-mini | 361.4 | 521.8 | 672.9 |

Table 10: Average response length of succinct, normal, verbose settings.

| Model Name/Running Type | Succinct | Normal | Verbose |
| --- | --- | --- | --- |
| GPT-3.5-Turbo | 13.04% | 14.13% | 13.04% |
| GPT-4o-mini | 20.65% | 20.65% | 19.57% |

Table 11: Accuracy of succinct, normal, verbose settings.

As can be seen, the difference of accuracy is at most $\approx 1\%$ for various response lengths, indicating that there is no significant length bias with our Evaluator.

### A.2.5 PROMPT DESIGN OF HINTS WRITER

The Hints Writer is prompted with the following instructions to generate three level hints for the issue, with `<ISSUE>` being the placeholder.

```
You are an issue-hint writer. You need to formulate some hints for
    detecting certain issues, based on the description in
<issue> below.

<rules>
- The hints should have three different levels, from very general to very
     specific.
- For ''general hint'', you should only tell the general issue, without
    any specific information.
```

```
- For ``medium hint'', you could tell which files are involved in this
    issue + general hint. Make sure it contains the information from the
    general hint.
- For ``specific hint'', you should tell which files are involved in this
    issue clearly, as well as a part of context information. Make sure
    it contains the information from the general and medium hints.
- You should never tell the agent the full context of the issue, only
    give it hints, so we can test to what degree the agent detects the
    issue and tell us evidence by themselves.
- Your word used in hints should be precise, pay attention to the word
    choice, and try to avoid misleading or ambiguous words. You should
    make sure you have a clear understanding of the issue before writing
    the hints.
- The hints should be clear and concise, try to cover all the issues
    mentioned and avoid unnecessary information.
</rules>

<example>
{
    <issue>
    {
        title: What is the provenance of these benchmark datasets?}
        content:
            {
                Firstly, check the English proverbs dataset: https://github.
                    com/google/BIG-bench/tree/main/bigbench/benchmark_tasks/
                    english_proverbs
                ```
                    task can help to draw conclusions about human-like
                        understanding and
                    ## Data source
                    ## References
                ```
                There is nothing in the data source section.
            }
        involved: {
            name: README.md,
            context: {
                ```
                    task can help to draw conclusions about human-like
                        understanding and Data Source References
                ```
            }
        }
    }
    </issue>

    output:
    {
        general: "section with empty content"
        medium: "a section with empty content in markdown file"
        specific: "a sub-heading section in README.md with empty content"
    }

</example>

------

Now let's begin

<issue>
<ISSUE>
</issue>

write your hints here in JSON format:
```

When designing the prompt, we take into consideration the risk that hints could provide too much information that can already serve as a valid answer. In the "<rule>" section, we clearly require the Hint Writer not to reveal the full context of the issue to the Curator, in case it takes it as a shortcut and skips reading the real files. We also manually check the generated hints to ensure the generated hints comply this rule.

## A.3 Detailed Analysis of the Performance of Baseline Curator on DCA-Bench

### A.3.1 From the Perspective of Different Content Length

Given the large average content length (or size of files, counted by token number) involved in test cases, it's interesting to explore how the success rate of the Baseline Curator changes with the number of tokens. Therefore, we created a statistical plot showing the count of success and failure cases for different content lengths. Fig. 4 shows that the success rate doesn't monotonically decrease as content length grows. For our baseline Curators, we used the OpenAI Assistant API with the Code Interpreter Tool to build the agents. This tool provides temporary file storage for uploaded files and can process them with code scripts. By processing data before it is passed to the LLM, the tool allows the OpenAI Assistant to handle much smaller inputs in terms of token usage, even when working with large datasets. Consequently, the effective input size for the LLM is often significantly reduced. Notably, depending on the agent's capacity to efficiently process and manage information.

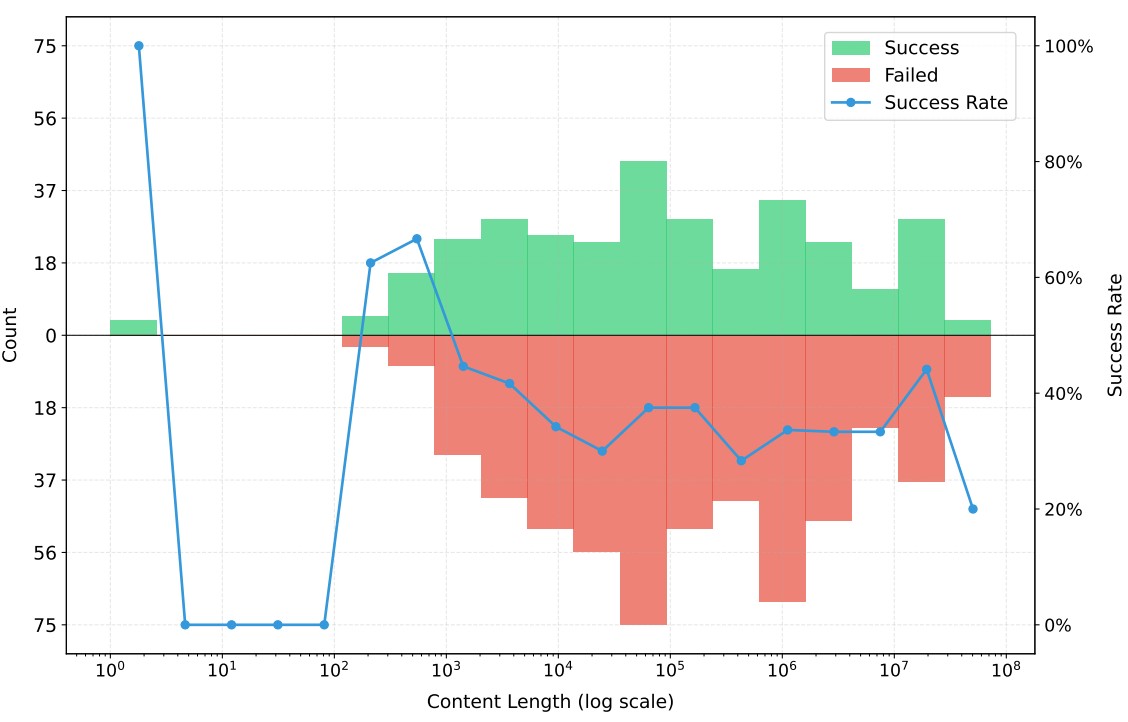

Figure 4: Number of success & failure and success rate versus different content length of test cases.

Besides, the performance of the Curator also depends on the type of issues. Take the issue 21ca944c-cf82-4764-bb2c-4c8db0cee950[11] as an example. This issue involves many missing values in a large CSV file. Even with 361,814 tokens, it's relatively easy for an LLM agent to spot the problem. The agent can quickly check part of the file and identify the missing values. This example shows that task difficulty isn't just about data size. It also depends on the problem type and how relevant the information is.

---

[11]https://www.kaggle.com/datasets/roche-data-science-coalition/uncover/discussion/168946

Therefore, it's expected that a test case with larger content length does not necessarily lead to poorer performance.

### A.3.2 FROM THE PERSPECTIVE OF HINT LEVEL

Here is a more detailed analysis of the overall performance with different hints. As outlined in L183-L186, the hints are divided into four levels, from 0 to 3, focusing on two dimensions with increasing information: the description of the issue and its specific context. The design follows an incremental pattern from h0 to h3. Please refer to Appendix 10 for some specific examples. Let's continue by discussing the capabilities required to succeed at each hint level.

If we aim to develop an LLM agent for this task without considering cost constraints, the process should involve the following steps: **Retrieve Knowledge → Plan and Select Files to Check → Read Files and Validate Issues → Receive Feedback and Update State → (Loop) → Make Final Decisions**. Since our Baseline Curator does not support looping, we exclude it from the following analysis. We will use this diagram to demonstrate which steps become less critical as the hint level increases.

**h0:** With no hints provided, the Curator must examine all the uploaded files. This requires the ability to handle and retain very long inputs effectively. To identify potential issues, the Curator needs comprehensive knowledge of dataset curation to anticipate a wide range of problems and retrieve relevant information when provided with the dataset files. The task also involves exploring and interacting with the dataset files to gradually narrow down the search space, requiring high-level reasoning and planning abilities. This corresponds to no reduction in the original diagram's required steps.

**h1:** When given a general description of the issues but no specific location, the search space is somewhat reduced but not entirely, as the description remains broad. A robust understanding of dataset curation and the ability to manage long inputs are still necessary. Ideally, this follows the pattern of Retrieval of Knowledge (↓), with minimal impact on the steps for planning and selecting files, except where file names are relevant. The steps for reading files and validating issues also see a slight reduction (↓).

**h2:** At this level, the Curator is provided with both a general description of the issue and specific hints about which files are involved. This further reduces the search space, making the ability to manage long inputs sufficient but less critical. The need for extensive knowledge retrieval decreases as the Curator can focus on the relevant files. Planning and file selection are simplified, as the Curator has clear guidance on which files to examine. Reading and validating issues also become easier, as the scope is narrower, reducing the overall complexity of the task. This follows the pattern of Retrieval of Knowledge (↓), Planning and Choosing Files (↓), and Reading and Validating Issues (↓).

**h3:** At the highest hint level, the Curator receives the general description, the specific files involved, and partial context. This greatly reduces the need for managing long inputs and makes the retrieval of knowledge more targeted. Planning and file selection is almost straightforward, with the Curator essentially guided to the areas needing attention. Reading and validating issues are significantly simplified due to the detailed context, further decreasing the Curator's workload. This follows the pattern of Retrieval of Knowledge (↓ ↓), Planning and Choosing Files (↓ ↓), and Reading and Validating Issues (↓ ↓).

### A.3.3 FROM THE PERSPECTIVE OF DIFFERENT ISSUE TAGS

We show the evaluation results from the Baseline Curator on the whole DCA-Bench classified into different tags in Tab. 12. We conduct analysis for some tags with relatively low success rates below.

**Typo** This refers to issues caused by typographical errors, including misspelled words, misused phrases, and incorrect email addresses. The specific nature of these issues is that they involve small errors within a long context, which means a large search space. This makes errors that are easy to spot in shorter contexts extremely difficult to find, even for humans.

Table 12: The success rate of Baseline Curator on DCA-Bench (221 samples × 4 hint levels) grouped by tags.

| Category | Success Rate/Number | Sub-category | Success Rate/Number |
|---|---|---|---|
| **typo** | **33.33% / 18*4** | — | — |
| wrong-format | 46.43% / 14*4 | — | — |
| inappropriate-file | 50.00% / 4*4 | — | — |
| ethical/legal-risk | 47.50% / 10*4 | — | — |
| **internal-discrepancy** | **36.90% / 21*4** | — | — |
| **cross-file-discrepancy** | **28.41% / 44*4** | — | — |
| data-problem | 32.99% / 197*4 | wrong-value | 29.58% / 71*4 |
| | | missing-value | 40.00% / 15*4 |
| | | data-leakage | 50.00% / 2*4 |
| | | apparent-corruption | 45.63% / 40*4 |
| | | hidden-corruption | 25.42% / 59*4 |
| document-problem | 40.96% / 83*4 | **wrong-info** | **29.63% / 27*4** |
| | | insufficient-info | 45.67% / 52*4 |
| infrastructure-problem | 35.53% / 19*4 | data-access | 56.25% / 4*4 |
| | | **script-code** | **30.00% / 15*4** |

**Internal-discrepancy**  This involves issues within a file where information is misaligned, such as a mismatched sum for some features in a .csv file[12], or a word misused in a way that contradicts itself within its context[13]. These issues require reasoning and memory over a long context.

**Cross-file-discrepancy**  Compared to internal-discrepancy, this type of issue requires checking for consistency across different files, such as misaligned information between a datacard and dataset files[14]. This can be challenging when there are many issues and limited hints about the files involved.

**Wrong-info**  This involves incorrect information in documentation files (e.g., wrong meta-information, typos, invalid URLs, and email addresses). This category overlaps with the issues mentioned above.

**Script-code**  Many datasets also provide script code to load the data, and issues can arise in these scripts, leading to data loading problems[15]. To address these issues, the Curator needs to be skilled in coding and debugging and sometimes must understand the dataset's purpose[16].

## A.4  ETHICS STATEMENT

### A.4.1  RESEARCH INVOLVING HUMAN PARTICIPANTS

The human annotators are the authors of this work, and we have confirmed that they are willing to annotate, knowing the potential ethically problematic data issues, as we listed in **Offensive Content**. Prior to the annotation, the leading authors have checked that the datasets with ethical problems pose minimal risks to the annotators.

---

[12]https://github.com/fivethirtyeight/data/issues/52
[13]https://github.com/google/BIG-bench/pull/904/files#diff-a5582cf9c7731f66cc3e8f6aa0e9d070483fc14f7c1d3545f275207c6fa39bd5
[14]https://github.com/fivethirtyeight/data/issues/261
[15]https://huggingface.co/datasets/spyysalo/bc2gm_corpus/discussions/3
[16]https://github.com/google/BIG-bench/pull/559

### A.4.2 DATA PRIVACY, COPYRIGHT, AND CONSENT

The DCA-Bench has collected materials and datasets from the following platforms: HuggingFace, Tensorflow Dataset, Graph Learning Indexer, Kaggle, BIG-bench, OpenML, Open-Data-Registry, and FiveThirtyEight. **The corresponding terms or license links are provided.**

Besides, **we have also included a detailed table `DCA_Benchmark.csv` in our supplementary materials.**

**where all the files involved in DCA-Bench have been annotated with their License Information.**

Each data point in DCA-Bench has two types of license:

- License for the platform that hosted this dataset
- License of this dataset

**Details:**

- Some data points in DCA-Bench involve files from 2 datasets, we listed all licenses.
- Many datasets themselves don't have a license listed on their data card.
- We notice that there are some datasets[17] claiming that "Data files © Original Authors", which is not a standard License. We have reached out to the dataset owners for clarification, but have not yet received a response. Therefore, we choose to record them as it claimed in our table.

**How does this secondary usage of user-generated data comply with restrictions?** DCA-Bench involves user-generated data (comments, modified codes) collected from dataset repositories hosted on GitHub, HuggingFace, and Kaggle.

**For GitHub**, we collected the comments and modified codes generated by users. According to section D.3, paragraph 2 of GitHub Terms of Service[18],

> "Because you retain ownership of and responsibility for Your Content, we need you to grant us—and other GitHub Users—certain legal permissions, listed in Sections D.4—D.7."

According to section D.5. License Grant to Other Users[19], if not provided a specific license, any User-Generated Content you post publicly, including issues, comments, and contributions to other Users' repositories, may be viewed by others. However, it doesn't clearly explain how this content is allowed to be used in which ways, and which is not. However, according to a Github issue discussion[20], we believe our practice is acceptable. Besides, we noticed that there have already been some works (Kocetkov et al., 2022; Jimenez et al., 2024b) with similar usage of GitHub data, which implies that this usage is acceptable.

**For HuggingFace**, according to HuggingFace Content Policy[21], Content types may include:

- "ML Artifacts": Code and assets hosted as Hugging Face Repositories, including Models, Datasets, and Spaces;
- "Community Content": Content that can be found in the Community section of the Hugging Face Platform, including discussions, comments, and usernames, as well as related documentation such as READMEs, model cards, data cards, pull requests, and merges.

According to HuggingFace Terms of Service[22], Section "Your Content",

---

[17] e.g. https://www.kaggle.com/datasets/roche-data-science-coalition/uncover/data

[18] https://docs.github.com/en/site-policy/github-terms/github-terms-of-service#c-acceptable-use

[19] https://docs.github.com/en/site-policy/github-terms/github-terms-of-service#5-license-grant-to-other-users

[20] https://github.com/orgs/community/discussions/135466

[21] https://huggingface.co/content-guidelines

[22] https://huggingface.co/terms-of-service

> "If you decide to set your Repository public, you grant each User a perpetual, irrevocable, worldwide, royalty-free, non-exclusive license to use, display, publish, reproduce, distribute, and make derivative works of your Content through our Services and functionalities;"

Therefore, we believe our usage of user-generated content from HuggingFace should belong to the "derivative works" here, which is acceptable.

**For Kaggle**, after reviewing their Terms of Use[23], we noticed that there are no explicit guidelines regarding the use of user-submitted content for academic research purposes. So we have sent Kaggle Support a request for clarification, and here is the response:

> " Kaggle is a neutral platform in hosting competitions and data. Protections under copyright law can vary based on the nature of data, contractual obligations, and your jurisdiction. As a result, we are not able to approve individual uses of data on a case-by-case basis. Copyright or usage questions should be directed to your legal counsel. You may post usage enquiries in the respective competition's forum, but we can not guarantee an official response from the Sponsor.
>
> If the rules do not specify usage restrictions, we ask that you use your best judgement in determining fair use. For example, it may be okay for you to use the data for teaching a class, demoing a method in a blog post, or other non-commercial purposes. Some of our competitions identify themselves as open-source and are meant to foster research, collaboration, and continued analysis by the data science community. "

Since our paper is not for commercial usage, we believe our practice is acceptable

Lastly, our collection and evaluation processes **exclude the gathering of any user information**. We promise that we **will remove the content once requested with a valid reason.**

### A.4.3 OFFENSIVE CONTENT

Our dataset, comprising 221 data instances, includes content that may be considered sensitive or potentially controversial. Specifically, there are 10 data points involving ethical or legal risks, among which 2 data points[24] contain content that may exhibit bias towards specific groups of people. Other issues are relevant to data source or license information.

Besides, the risk in most cases is associated with deploying models trained with these datasets. Reviewing the quality issues of these datasets poses minimal risks to the annotators.

**Justification for Inclusion**

While we acknowledge the sensitive nature of these data instances, their inclusion in our dataset is both intentional and necessary for the following reasons:

- **Benchmark Objectives**: One of the primary goals of our benchmark is to examine LLM agents' ability to identify and assess potential ethical and legal risks in AI training data. The inclusion of these sensitive data points is crucial for thoroughly evaluating the capability of AI models to recognize and appropriately handle such content.

- **Realistic Representation**: These data points reflect real-world scenarios that AI systems may encounter. By including them, we ensure our benchmark provides a more comprehensive and authentic assessment of AI performance.

---

[23]https://www.kaggle.com/terms

[24]https://github.com/google/BIG-bench/pull/685 and https://www.kaggle.com/datasets/vikrishnan/boston-house-prices/discussion/429030

## A.5 OTHER TECHNICAL DETAILS

### A.5.1 DISCUSSION ON THE DIFFERENCE BETWEEN OPENAI ASSISTANT API AND CHATGPT

The performance discrepancy between OpenAI's Assistant API and ChatGPT has been widely discussed within the OpenAI user community. Users often report that the Assistant API's performance does not match the capabilities of ChatGPT, especially in tasks involving complex file processing and analysis.

The Assistant API provides several methods for handling files: 1. Code Interpreter for file uploads. 2. Building VectorStores via File Search. 3. Attaching files to messages using their file IDs.

However, these methods have significant issues: - The quality of the VectorStore built using File Search is inferior to ChatGPT's built-in file search capabilities. - Files are referenced by their file IDs rather than their names, leading to confusion and errors during processing.

The following code snippet illustrates the process of uploading and referencing files in the Assistant API by attaching them to the messages:

```python
msg_file_ids = []
for file_path in file_paths:
    message_file = self.client.files.create(
        file=open(file_path, "rb"), purpose="assistants"
    )
    msg_file_ids.append(message_file.id)

attachments = [{"id": file_id, "tools": [{"type": "file_search"}]} for
    file_id in msg_file_ids]

thread = self.client.beta.threads.create(
    messages=[
        {
            "role": "user",
            "content": input,
            "attachments": attachments
        }
    ]
)
```

Despite correctly assigning file names during the upload process, the Assistant API seems to ignore these names during subsequent file processing, relying solely on file IDs. This results in frequent mix-ups and incorrect file references.

Several forum posts and articles highlight deficiencies relevant to this topic: - Different responses assistant playground vs API - using the same assistant ID[25] - Huge difference between ChatGPT assistant and API assistant[26] - Assistant API not consistent[27]. This implies the limitations of OpenAI Assistant API.

In conclusion, while the Assistant API offers various tools for convenient file handling, its current implementation presents significant challenges that hinder its effectiveness compared to ChatGPT, which motivates us to select a subset of DCA-Bench containing hard cases to test on ChatGPT.

### A.5.2 CALCULATION OF FILE NUMBER AND CONTEXT LENGTH

We calculate the file context length using the `tiktoken` package[28] to tokenize the file content, using:

```python
import tiktoken
```

---

[25]https://community.openai.com/t/different-responses-assistant-playground-vs-api-using-same-assistant-id/547562

[26]https://community.openai.com/t/huge-difference-between-chatgpt-assistant-and-api-assistant/587335

[27]https://community.openai.com/t/assistant-api-not-consistant/718145

[28]https://github.com/openai/tiktoken

```
def calculate_token_len(sentence, model="gpt-4-0125-preview"):
    encoding = tiktoken.encoding_for_model(model)
    tokens = encoding.encode(sentence)
    token_len = len(tokens)
    return token_len
```

During calculation, non-text files are skipped, which include .jpg, .jpeg, .png, .gif, .bmp, .mp3, .wav, .ogg, .flac, .mp4, .avi, .mov, .mkv.

For .zip files, we will unzip them to calculate the context length of text files. However, when calculating the file number, we take the whole .zip number as one single file.

### A.5.3 COST ANALYSIS

In this section, we provide an estimated cost analysis for applying our Evaluator backend by `GPT-4-0125-preview` to rate the performance of the Baseline Curator (OpenAI Assistant API) on DCA-Bench (884 inputs in total). We applied `tiktoken` for tokenization of contexts.

Based on the pricing information available as of May 2024 from https://openai.com/api/pricing/, the costs are: input cost: $16.12, output cost: $28.99, total cost: $45.11.

The costs are rounded to two decimal places. Note the cost here only involves the Evaluator; we dismiss the cost to run the baseline Curator on DCA-Bench to get the outputs, which depends on the design of the Curator and thus is not our focus.

