# OpenReview forum: "DCA-Bench: A Benchmark for Dataset Curation Agents"
_ICLR.cc/2025/Conference — Submitted to ICLR 2025_

### Official Review · Reviewer_WL5u · 2024-11-02

**Soundness:** 2
**Presentation:** 3
**Contribution:** 3
**Rating:** 5
**Confidence:** 3

**Summary:**

This paper introduces a novel benchmark for assessing the capability of LM-based curator agents in identifying potential issues within datasets. The authors also streamline the evaluation process, enabling non-expert usage. Baseline evaluations conducted with GPT-4 demonstrate the system’s performance and highlight the potential value of their autonomous dataset curation approach.

**Strengths:**

1. This paper establishes a complete curation agent benchmark derived from real-world datasets, providing test cases ranging from easy to challenging, and potentially can be serve as an important foundation in this area.
2. The paper is well-organized, presenting a clear workflow of the proposed method alongside some demonstrations that effectively validate the robustness of the evaluation framework.

**Weaknesses:**

From my perspective, this paper presents the following limitations:
1. The benchmark appears to rely primarily on data sources from machine learning engineering, with a limited number of test cases. This raises questions about its generalizability to other domains, such as image datasets.
2. While the work presents a well-defined benchmark with a structured testing procedure, it largely resembles a series of experiments conducted with GPT-4 rather than a comprehensive agent-testing framework. The experiments on more agents are insufficiently thorough.
3. The paper does not provide open-source code.

**Questions:**

Can your method be effectively applied to other domains, such as image datasets like ImageNet? Do you intend to make the benchmark publicly available?\
\
\
Additional: \
Pointing out an issue with your paper’s formatting: typically, one page contains around 53 lines of content, but your paper only has 46 lines, like page 1.

---

> ### Author Response · Authors · 2024-11-21
>
> ### 1. Regarding Domain Scopes
>
> Our DCA-Bench is an initial step toward this important topic and aims to provide some preliminary insights. A more detailed analysis of domain-specific data with unique properties would undoubtedly be valuable but might require deeper domain-specific knowledge and more refined designs. Modalities beyond text are likely to introduce additional challenges. **However, we believe the core components and frameworks of the benchmark—such as issue description, issue context, and ground truth collected from user reports—can be adapted to other domains and modalities.**
>
> Incorporating domain-specific or multi-modal content would make issue detection even more challenging, emphasizing the need for carefully designed hints to enable more precise evaluation. Our LLM Evaluator could also be replaced with tools integrating multi-modal or domain-specific knowledge, thereby enhancing its ability to assess the performance of LLM agents.
>
> **In conclusion, our approach has the potential to generalize to other domains, such as image datasets.** We aim to explore these possibilities further in future work.
>
>
> ### 2. Regarding Agent-Testing Framework
>
> We would like to clarify that our baseline Curators are already implemented within an agent framework. The experiments for the Baseline Curators are based on the OpenAI Assistant API, which provides tools including a code interpreter, file processing, and storage.  Therefore, the baseline Curators evaluated in DCA-Bench are LLM agents with certain tool usage, as opposed to pure LLMs.
>
> We envision that considering more tools and introducing more agents for **collaborative problem-solving (as mentioned on the left of Fig. 2) are meaningful and exciting next steps.** The API design of DCA-Bench can accommodate the evaluation of more complicated Curator agents, as we have taken these into consideration.
>
> ### 3. Availability of Open Source Code
>
> According to the ICLR 2025 Author Guide, we cannot provide any URL that might reveal our identity, so **we uploaded all of our code as supplementary materials in our submission**, where we also provided information about how to download our dataset. Please refer to the “Reproduction Guidelines” in the README file in the supplementary materials for more details.
>
>
> ### 4. Applicability to Datasets like ImageNet
>
> Thank you for raising this point! Please see our response to point 1 for more details.
>
> ### 5. Plan to Publicize Our Benchmark
>
> Yes, we will release our benchmark code and dataset once we get accepted.
>
> ### 6. Format Issue
>
> We appreciate you bringing this to our attention! The formatting issue has been addressed in the updated draft.

---

### Official Review · Reviewer_VZxc · 2024-11-04

**Soundness:** 2
**Presentation:** 2
**Contribution:** 3
**Rating:** 6
**Confidence:** 2

**Summary:**

The paper introduces a Dataset Curation Agents Benchmark to evaluate the ability of LLM agents to identify hidden data quality issues in community-contributed datasets. Focusing on the challenge of detecting undocumented issues rather than solving known problems, this benchmark uses 221 real-world test cases from popular dataset platforms and an automatic evaluation framework utilizing GPT-4, with alignment to expert evaluations. Initial results indicate a baseline GPT-4 Curator detects only a small percentage of issues, underscoring the complexity and need for further research in autonomous dataset curation.

**Strengths:**

1. Innovative Concept: The paper presents a novel approach by shifting focus from issue-solving to issue-discovery in dataset curation.
2. Comprehensive Coverage: DCA-Bench provides a broad testing ground with diverse curated test cases, representing real-world scenarios across different platforms.
3. Automatic Evaluation Framework: The use of GPT-4 for evaluation is a practical advancement, offering scalability where human annotation is not feasible.
4. Clear Practical Relevance: Tackling hidden data quality issues addresses a significant gap in dataset management, which is crucial for improving AI research outcomes.

**Weaknesses:**

1.	Limited Test Set Size: With only 221 samples, and limited instances in some categories (e.g., only 10 ethical instances), the test set might not sufficiently capture the complexity of real-world data quality issues. Increasing the dataset size and diversity could improve robustness.
2.	Missing Related Works: The paper does not adequately engage with existing literature regarding data system and LLM agents, such as [arXiv.2402.02643, LLM-Enhanced Data Management], [SIGMOD’24, Data-juicer: A one-stop data processing system for large language models], and [ICML'24, DS-Agent: Automated Data Science by Empowering Large Language Models with Case-Based Reasoning], which could provide valuable context and highlight the novelty of this work.
3.	Clarity Issues and minor writing suggestions: There are several presentation issues, such as missing references to Figure 1 in the text and unclear definitions in Table 1 for sample-level, type-level, and tag-level insights. Addressing these could enhance readability and comprehension. Beside, Line 104-105 lacks a “:” Line 213~214 lacks the references of “Kaggle、OpenML、TF-Dataset、Open-Data-Registry, Five-Thirty-Eight”.
4.	Applicability to different types of datasets: only eight open data set platforms have been mentioned, but there is a lack of analysis on whether different types of data sets (such as medical data, financial data, etc., have special properties) are equally applicable. It is recommended to add discussion or preliminary experimental results to demonstrate the versatility of DCA-Bench.

**Questions:**

None, plz see the weakness above

---

> ### Author Response · Authors · 2024-11-21
>
> ### 1. Limited test set size
>
> Many well-established LLM benchmarks have sample sizes at hundred levels: HumanEval [1] (164), MTBench [2] (80), and ArenaHard [3] (500). As an initial step toward the novel direction of automatic data curation, we believe the size of the current test set is sufficient for a robust evaluation. **In addition, DCA-Bench features a 4-level hints design, resulting in actually 884 test cases in total.** However, we recognize that increasing the dataset size could further enhance robustness and improve the balance of test cases across different categories, which is an important future work.
>
> \[1\] M. Chen et al., “Evaluating Large Language Models Trained on Code,” Jul. 14, 2021, arXiv: arXiv:2107.03374. doi: 10.48550/arXiv.2107.03374.
>
> \[2\] L. Zheng et al., “Judging LLM-as-a-Judge with MT-Bench and Chatbot Arena,” NeurIPS 2023
>
> \[3\] T. Li et al., “From Crowdsourced Data to High-Quality Benchmarks: Arena-Hard and BenchBuilder Pipeline,” Jun. 17, 2024, arXiv: arXiv:2406.11939. doi: 10.48550/arXiv.2406.11939.
>
> ### 2. About Related Works
>
> We appreciate your suggestions on the related works. These works are all innovative and meaningful; while they primarily focus on improving data management, processing, or problem-solving frameworks, **our work is distinguished by its focus on identifying hidden issues in public datasets**, such as factual or logical errors that require semantic understanding, and usability aspects like document quality and licensing—areas that are critical but often overlooked in related research. The detailed discussions of these relevant works are as follows:
>
> - LLM-Enhanced Data Management [1] designed and implemented a data management system that applied LLMs for data management tasks, such as database diagnosis and data analytics, and also shows the potential of using such a data system to enhance the performance of LLMs. However, it is more relevant to database management where LLMs are asked to retrieve data for analytics and reasoning or reorganize data using a series of APIs and tools. **This work didn't target detecting the issues within a single dataset, highlighting a different task scope between DCA-Bench and this work.**
> - Data Juicer [2] serves as a comprehensive data processing system for LLM training, which provides numerous operators that can be used to generate a high-quality dataset from a mixture of data sources. However, the data quality enhancers in Data Juicer are rule- or metrics-based filters or deduplicators, **which cannot handle issues** like factual errors or logical errors that require semantic understanding—what we refer to as "hidden corruptions" in DCA-Bench (please refer to Section A1.2 for more details). Besides, **it also didn't focus on dataset-level issues**, where usability—including the quality of the document and the presence of licenses—matters a lot. This highlights the novelty of the tasks in DCA-Bench.
> - DS-Agent [3] proposes an agentic framework for solving data science problems, such as Kaggle competitions. While this work leverages LLMs for data science tasks, it primarily aims to develop a problem-solving agent framework. In contrast, **DCA-Bench has different motivations and focuses on identifying and addressing dataset issues rather than developing agents for data science.**
>
> **We have integrated these discussions into the revised draft, marked with blue under “Related Works”.**
>
> \[1\] Zhou, Xuanhe, Xinyang Zhao, and Guoliang Li. "LLM-Enhanced Data Management." arXiv preprint arXiv:2402.02643 (2024).
>
> \[2\] Chen, Daoyuan, et al. "Data-juicer: A one-stop data processing system for large language models." Companion of the 2024 International Conference on Management of Data. 2024.
>
> \[3\] Guo, Siyuan, et al. "DS-Agent: Automated Data Science by Empowering Large Language Models with Case-Based Reasoning." arXiv preprint arXiv:2402.17453 (2024).
>
> ### 3. About Writings
>
> Thank you for your thorough review and feedback on writing!  We have addressed all of them in our latest draft. **For easier comparison, we have marked these changes in blue.**
>
>
> ### 4. Applicability to different types of datasets:
>
> Our DCA-Bench is an initial step toward this important topic, and we intend to provide initial insights. More detailed analyses of domain-specific data with special properties are meaningful but should need more domain-specific knowledge and more refined designs, which we leave for future works. Currently, we classify those issues **according to the type of issues themselves rather than domains**, where we provide clarifications and analysis regarding some challenging issue types, such as "cross-file discrepancy" where the agent needs to spot discrepancies of the same information presented in multiple files. **Please see A3.3 for more details.**

---

### Official Review · Reviewer_joVR · 2024-11-05

**Soundness:** 3
**Presentation:** 3
**Contribution:** 2
**Rating:** 5
**Confidence:** 2

**Summary:**

The work introduces a benchmark aimed at evaluating the ability of large language model (LLM) agents to autonomously detect subtle data quality issues in large, real-world datasets.

The work addresses the ongoing challenges in maintaining data quality on open platforms, where issues like mislabeling, documentation gaps, and ethical concerns are prevalent.

DCA-Bench comprises 221 real-world test cases across eight major dataset platforms and employs an automatic evaluation framework using GPT-4 to assess the performance of LLMs as dataset curators.

**Strengths:**

- The work addresses a significant problem in AI, as dataset quality is critical for robust model performance and reliable research outcomes.
- The work provides a foundation for future research into fully autonomous dataset curation systems, which could save time and reduce errors in data management.
- The framework offers different hint levels to evaluate LLMs at multiple stages of problem discovery, providing nuanced insights into model capabilities.
- DCA-Bench constructs an automatic evaluation pipeline: leverages GPT-4 to automate evaluations, which aligns well with expert human assessments, making the process scalable and consistent.

**Weaknesses:**

- There is a lack of a fully realistic testing environment. While comprehensive, the test cases may not fully represent the diversity of data quality issues encountered in real-world curation.
- Performance is heavily dependent on hints. Without hints, the baseline LLM agent detected only 11% of issues, indicating a limited ability to autonomously identify dataset problems.
- The automatic evaluation pipeline, while effective, may introduce subtle biases that differ from nuanced human judgment.

**Questions:**

The benchmark is primarily text-focused, will the benchmark be scalable to multimodal data (e.g., image or audio datasets)?

---

> ### Author Response · Authors · 2024-11-21
>
> ### 1. About Realistic Testing Environment and Coverage of Issue Cases
>
> We agree with the reviewer that a more realistic testing environment can further help develop and test candidate models. At the current stage, we chose to simplify the setup to focus on common concepts. Regarding comprehensiveness, our DCA-Bench serves as an initial step in this direction. While it may not yet cover all real-world issue types, it already provides a very challenging and meaningful testbed, as demonstrated by our experiments. Moreover, its structured taxonomy of issue types and tags, detailed in Sections A.3.2 and A.3.3, provides further depth and utility.
>
> ### 2. Hints Design and the Performance of Baseline Agent
>
> - We want to clarify that the design of multi-level hints is a feature of DCA-Bench; **we didn’t mean to use it to improve the performance of the baseline agent**. With credit to this design, we carried out a more detailed analysis of the overall performance with different hints in Section A.3.2.
> - **It is not uncommon that the baseline methods have relatively low performance in new LLM Benchmarks [1, 2] that target intrinsically difficult tasks.** The dataset issue detection task is quite challenging for LLM agents, requiring them to reason under a very long context, as shown in Tab.1. We have also analyzed the challenges posed by the issues in DCA-Bench, categorizing them by issue tags as detailed in Section A.3.3.
>
> [1]: Jimenez C E, Yang J, Wettig A, et al. Swe-bench: Can language models resolve real-world github issues? ICLR 2024
>
> [2]: T. Li et al., “From Crowdsourced Data to High-Quality Benchmarks: Arena-Hard and BenchBuilder Pipeline,” Jun. 17, 2024, arXiv: arXiv:2406.11939. doi: 10.48550/arXiv.2406.11939.
>
>
> ### 3. Potential biases of LLM Evaluators
>
> In Lines 379-381(Lines 390-393, in the revised version), we have considered two common biases among LLMs: self-preference bias [1, 2] and length bias [3,4]. In **Section A. 2.4**, we have carried out well-designed experiments for both of them. **Our results showed that these biases are negligible in our LLM Evaluator.**
>
> - **For the self-preference experiment**, we used GPT-3.5-Turbo and GPT-4-0125-preview as the backend models for both the Curators and the Evaluators, leading to four combinations of Curator and Evaluator settings. For each Evaluator, we calculated the Average Odds Difference (AOD), a metric that measures fairness by assessing differences in true positive and false positive rates between two groups. A value of zero indicates perfect fairness, with equal model performance across both groups. Higher AOD values suggest increasing disparity, with the model performing unevenly between the groups. Derived from Tab.7 and Tab.8, we showed that our GPT-4-0125-preview based Evaluator has an acceptable AOD ($ 5.31(\pm2.76)\\% < 10\\%$), which indicates the bias is within an acceptable range [5]. Furthermore, we observed a **great alignment between the GPT-4-0125-preview based Evaluator and human annotations**, on both the GPT-3.5-Turbo based Curator and the GPT-4-0125-preview based Curator, as shown in Tab.9.
> - **For length bias**, we used GPT-3.5-Turbo (0125) and GPT-4o-mini (0718) as the dataset curator agents. After the Curator generated the results, we fed its response to another LLM (gpt-4o-2024-08-06) to expand upon its generation. **While the most common bias is the preference towards longer responses, we also tested the case where the response is more succinct.** As shown in Tab.10 and Tab.11 of our draft, the changes in the success rate given by the LLM Evaluator are negligible in both cases, indicating that our Evaluator has no significant length bias.
>
> \[1\] A. Panickssery, S. R. Bowman, and S. Feng, “LLM Evaluators Recognize and Favor Their Own Generations,” Apr. 15, 2024, arXiv: arXiv:2404.13076. doi: 10.48550/arXiv.2404.13076.
>
> \[2\] L. Zheng et al., “Judging LLM-as-a-Judge with MT-Bench and Chatbot Arena,” NeurIPS 2023
>
> \[3\] Y. Dubois, B. Galambosi, P. Liang, and T. B. Hashimoto, “Length-Controlled AlpacaEval: A Simple Way to Debias Automatic Evaluators,” COLM 2024
>
> \[4\] E. Durmus, F. Ladhak, and T. Hashimoto, “Spurious Correlations in Reference-Free Evaluation of Text Generation,”  ACL 2024
>
> \[5\] S. Majumder, J. Chakraborty, G. R. Bai, K. T. Stolee, and T. Menzies, “Fair Enough: Searching for Sufficient Measures of Fairness,” Mar. 21, 2022, arXiv: arXiv:2110.13029. Accessed: Aug. 13, 2024\. \[Online\]. Available: [http://arxiv.org/abs/2110.13029](http://arxiv.org/abs/2110.13029).
>
> ### 4. Scale to Multimodal Data
>
> We appreciate the reviewer’s suggestion. While DCA-Bench is currently text-focused, we believe it can be scaled to multimodal data, such as image or audio datasets. However, this would require more refined designs to address the unique complexities of multimodal data quality issues. We consider this an important direction for future work.

---

### Official Review · Reviewer_pFuZ · 2024-11-11

**Soundness:** 3
**Presentation:** 3
**Contribution:** 3
**Rating:** 6
**Confidence:** 4

**Summary:**

This paper aims to solve the detection of data quality issues such as data errors, documentation issues, file discrepancies, and legal/ethical risks. This study of this paper starts from curating 221 test cases from eight popular dataset publishing platforms (like HuggingFace, Kaggle) and propose an automatic evaluation framework using GPT-4. It is interesting to know that without any hints, a baseline GPT-4 Curator agent can only reveal 11% of the data quality issues in the 221 test cases. When given the most specific hint, 70% of the issues can be detected.  This is understandable because finding the hidden issues of data quality is a complex task, even for human experts like dataset users and platform maintainers.

**Strengths:**

S1: the exploration of data quality issues is an interesting and important problem. If LLMs are able to automate the detection of data quality issues, they will be helpful for automatic maintenance of data publishing platform.
S2:  The dataset collection process seems reasonable. Four different types of issues are annotated, representing the typical issues that may happen in public data platforms.
S3: The evaluation system leverages LLMs in multiple perspectives for implementing different evaluation strategies, using RAG, code interpreter to write and execute programs, etc.

**Weaknesses:**

W1: the Evaluator is designed to leverage LLMs as judges for assessing whether the detection results are correct or not. However, as authors mentioned:  the annotated test data for the Evaluator are collected after the prompt design for the Evaluator. This raises concerns that the test data may simply align with the designed prompts, potentially indicating that the Evaluator’s performance is optimized for this specific data. This brings into question  about how  the proposed prompt design of the Evaluator can generalize across different Curators and issues in the DCA-Bench.

W2: The performance of the system in identifying dataset quality issues appears to have been evaluated solely on datasets that are known to have issues. It is currently unclear how the system performs on high-quality datasets where no issues are present. This raises concerns about its ability to avoid false positives in scenarios where no data issues exist.

**Questions:**

Are high-quality datasets included in the evaluation process? How about false positives (i.e., incorrect identification of issues in clean datasets) if using the proposed detection models.

---

> ### Author Response · Authors · 2024-11-20
>
> We thank the reviewer for the detailed and thoughtful comments. We address your individual concerns below.
>
> ### 1. Regarding Evaluation Details of the LLM Evaluator
>
> We want to clarify that the main point we want to deliver in L332 - L335 (L342 - L345 in the revised version) is that we didn’t optimize the prompt design of the LLM Evaluators on the test data used for Tab2. As we mentioned in Line 344 (L354 in the revised version), the test cases are **randomly selected**, thus we didn’t try to cherry-pick the test data to make the performance better. The reason that test data are randomly collected after the prompt design is precisely to avoid overfitting in the prompt design. **We have also included the labeled test data under `pipeline/label.json` in our uploaded supplementary materials for your reference.**
>
> ### 2. Taking High-quality Dataset into Consideration
>
> We sincerely appreciate the reviewer’s observation and the opportunity to address this important concern. Our focus on datasets reported to have issues is based on two key considerations: annotation challenges and practical utility.
> - **Annotation Challenges:** Labeling a dataset as “issue-free” is inherently challenging. Human experts tasked with identifying issues must thoroughly examine the dataset, yet subtle or latent issues may go undetected. Conversely, labeling a dataset as “containing an issue” is more straightforward, as a single verified anomaly suffices to confirm its status. This asymmetry in annotation reliability underpins our choice of evaluation datasets.
> - **Practical Utility:** From a dataset curator’s perspective, optimizing for high recall (i.e., identifying as many potential issues as possible) is generally more beneficial than maximizing precision. False positives, while not ideal, are relatively easy for a human maintainer to validate or dismiss based on the auxiliary evidence provided. This prioritization aligns with the practical needs of curators, where the cost of missed issues often outweighs the effort of validating flagged data.

---

> > ### Comment · Reviewer_pFuZ · 2024-11-26
> > **reply to rebuttal**
> >
> > Thanks for the reply to my questions.
> > I am still interested in the evaluation on "issue-free" cases, because it is important to investigate false positives (i.e., incorrect identification of issues in clean datasets).

---

> > > ### Author Response · Authors · 2024-11-29
> > >
> > > We agree that this is an interesting direction to explore. However, thoroughly evaluating this point would require significant effort to collect and annotate additional issues, as well as to conduct experiments on these new issues. Unfortunately, this is not feasible at this time. While such efforts could further enhance the benchmark, we believe the current benchmark already makes a valuable contribution.
> > >
> > > Specifically, it serves as an initial step toward the development of AI agents for dataset curation, shifting the focus from issue-solving to issue detection. We have collected 221 samples and designed a multi-hint framework to enable a more refined analysis of the Curator’s performance. Our experiments demonstrate that the task is already challenging.
> > > Given these considerations, we have chosen to leave the evaluation of “issue-free” cases as an important direction for future exploration.

---

### Author Response · Authors · 2024-11-25

Dear reviewers,

As the conclusion of our rebuttal period approaches, I wanted to kindly check if our rebuttal has addressed your concerns. We greatly value your thoughtful feedback and have incorporated clarifications and updates into our drafts accordingly.

If you have any additional concerns, please do not hesitate to let us know at your earliest convenience, so we can strive to respond before the discussion period concludes.

Thank you very much for your time and consideration.

---

### Meta-Review · Area_Chair_n18x · 2024-12-21

**Metareview:**

The paper introduces the DCA-Bench to evaluate the ability of large language model (LLM) agents to detect data quality issues. DCA-Bench constructs an automatic evaluation pipeline by leveraging GPT-4 for automation. This research focuses on a vital issue, and some experimental results provide valuable insights. However, the primary challenge lies in the ubiquitous nature of detecting data quality issues for LLMs. Given the scale and complexity of this problem, the small-sized DCA-Bench offers only limited contributions. To enhance its impact, further exploration of domain-specific data is necessary.

**Additional Comments On Reviewer Discussion:**

The authors address questions such as:

1. Add discussions on Related Works.
2. Clarify the design of hints and the evaluation pipeline.
3. Discuss the potential biases in the evaluation process.

While this work represents an initial and promising attempt, it falls short of being high-quality due to some limitations. The primary challenge lies in the ubiquitous nature of detecting data quality issues for LLMs. Given the scale and complexity of this problem, the small-sized DCA-Bench offers only limited contributions. To enhance its impact, further exploration of domain-specific data is necessary.

---

### Decision · Program_Chairs · 2025-01-22

Reject